# A Latent Multilayer Graphical Model for Complex, Interdependent Systems

**Martin Ondrus**
Neuroscience and Mental Health Institute
University of Alberta

**Ivor Cribben**
Alberta School of Business
University of Alberta

**Yang Feng**
School of Global Public Health
New York University

## Abstract

Networks have been extensively used and have provided novel insights across a wide variety of research areas. However, many real-world systems are, in fact, a "network of networks", or a multilayer network, which interact as components of a larger multimodal system. A major difficulty in this multilayer framework is the estimation of interlayer edges or connections. In this work, we propose a new estimation method, called multilayer sparse + low-rank inverse covariance estimation (multiSLICE), which estimates the interlayer edges. multiSLICE bridges latent variable Gaussian graphical methods with multilayer networks, offering a flexible framework for modeling processes with irregular sampling and heterogeneous graph structures. We develop an effective algorithm to compute the estimator. We also establish theoretical conditions for the recoverability of the joint space, analyze how interlayer interactions influence joint parameter estimation, and provide theoretical bounds on their relationships. Finally, we rigorously evaluate our method on both simulated and multimodal neuroimaging data, demonstrating improvements over state-of-the-art approaches. All the relevant R code implementing the method in the article is available on GitHub.

## 1 Introduction

Our world is a complex assemblage of many interdependent systems that can be described as sets of interacting components. Networks (or graphs, which we use interchangeably) succinctly represent these systems, wherein individual actors (nodes) are connected through relationships (edges) that often encode some quantitative meaning (Wasserman and Faust, 1994). By representing systems as graphs, local and global graph attributes can describe phenomena in social analysis (Wasserman and Faust, 1994), genomics (Seal et al., 2023; Argelaguet et al., 2018), and neuroscience (Bassett and Sporns, 2017; Bassett et al., 2011), among others.

Gaussian graphical models (GGMs) estimate dependencies in such systems by modeling them as multivariate Gaussian distributions, parameterized by the inverse covariance (precision) matrix (Hastie et al., 2009). However, many real-world systems are in fact a "network of networks" (Craven and Wellman, 1973) that interact as parts of a larger multimodal system. For example, a logistics network may contain multiple transportation types or "layers" (e.g., air, rail, pipeline, road), which have within (intralayer) and between (interlayer) relationships. In neuroscience, genes, cells, tissue, anatomical structure, and functional measures can interact and contribute to behaviors or pathologies. Multilayer graphs naturally extend to such cases and encode rich multimodal data into a unified representation (Kivelä et al., 2014), but learning the dependence structure remains difficult.

39th Conference on Neural Information Processing Systems (NeurIPS 2025).

In such systems, estimating the dependence across different modalities presents challenges due to differences in the number of measured variables, as well as the sampling rate. In neuroscience, for example, measurement can be made across multiple modalities, such as magneto/electroencephalography (M/EEG), functional magnetic resonance imaging (fMRI), and structural magnetic resonance imaging (sMRI), among others. Each modality has unique strengths and weaknesses that are typically complementary to each other. For example, M/EEG has high temporal resolution but poor spatial resolution, and vice versa for fMRI. Different layers are often tightly coupled, with structure affecting function and vice versa. The challenge in combining the information across these modalities is that each may differ in the number of variables (nodes) and/or the number of samples. Furthermore, these measurements are taken independently from one another, creating additional issues on how to best structure the "jointness" of these measurements. Ideally, we endeavor to preserve the natural structure of the data by avoiding up- or down-sampling between different modalities. We may wish to incorporate information from other variables not observed across all modalities. As such, there is a clear need for a flexible framework that brings together these multiple modalities.

To this end, we introduce multiSLICE, which bridges multilayer networks (Kivelä et al., 2014) and latent variable GGMs (Chandrasekaran et al., 2012) by providing a flexible framework for modeling processes with irregular sampling and heterogeneous graph structures. In our setting, we consider a sparse + low-rank setup, where each modality has a sparse component, and multiple modalities are allowed to exist over a joint low-rank, latent space. Crucially, in a neuroscience context, the sparse component captures local circuits between different regions of the brain, while the low-rank component depicts common environmental influences across modalities (Yatsenko et al., 2015). To our knowledge, this is the *first* statistical model for estimating interlayer edges in a multilayer network with different numbers of variables and sample sizes, and one that is scalable in the number of modalities it can integrate. Our main contributions can be summarized as follows:

- We link latent variable Gaussian graphical methods and multilayer networks, establishing a flexible framework to model processes with irregular sampling and non-identical nodes.
- We derive an effective algorithm to solve for this estimator.
- We establish theoretical conditions for the recoverability of the joint space, analyze how interlayer interactions influence joint parameter estimation, and provide theoretical bounds on their relationships.
- We rigorously test our method on both simulated and real experimental data, with comparisons to the state-of-the-art, supporting the efficacy of our approach in both settings.

## 2    Related methods

In general, related methods accommodate either differing numbers of variables ($p$) between layers or differing sample sizes ($n$). Table 1 provides a summary. Mohan et al. (2014) suggests a node-based method (CNJGL) which uses a row-column overlap norm based approach for estimation. Lin et al. (2016) propose a multilayer GGM via penalized likelihood estimation (MLGGM). Gan et al. (2019) uses a Bayesian group regularization method and spike-and-slab Lasso priors in the proposed BJEMGM method. Price et al. (2021) suggests a cluster fusion regularization (CFR) based method to estimate multiple precision matrices. The BANS method models hierarchical dependencies using Bayesian node-wise selection (Ha et al., 2021). JMMLE extends this method by decomposing the multilayer problem into two-layer subproblems using neighborhood selection and group-penalized regression (Majumdar and Michailidis, 2022). Chang et al. (2022) considers a graph quilting problem, in which observations of the covariance are missing, and proposes a matrix-completion-based method (LRGQ) to estimate the missing entries. Albanese et al. (2024) suggests a collaborative graphical lasso (coglasso) for the estimation of multi-omics network data.

| | CNJGL (2014) | MLGGM (2016) | BJEMGM (2019) | CFR (2021) | BANS (2021) | JMMLE (2022) | LRGQ (2022) | coglasso (2024) | multiSLICE (Ours) |
|---|---|---|---|---|---|---|---|---|---|
| $n$ | **Yes** | No | **Yes** | **Yes** | No | No | **Yes** | No | **Yes** |
| $p$ | No | **Yes** | No | No | **Yes** | **Yes** | **Yes** | **Yes** | **Yes** |

Table 1: A summary of methods in columns with year of publication. Entries indicate whether the approach can handle different sample sizes ($n$) or different node sets ($p$) across modalities.

# 3 Preliminaries

## 3.1 Notation

A graph $G$ is defined as a collection of vertices (or nodes) and edges, $G = (V, E)$. For the edge set of a weighted graph, we denote the edge between nodes $i$ and $j$ with weight $w_{ij}$ by the tuple $(i, j, w_{ij}) \in E$. Alternatively, $G$ can also be described by an adjacency matrix denoted by $\boldsymbol{A}$. The cardinality of a set $s$, or the number of elements in the set, is denoted by $|s|$. We denote a matrix $\boldsymbol{B}$ by a bold uppercase letter, a vector $\boldsymbol{b}$ with a bold lowercase letter, and a scalar $a$ with a lowercase letter. $\boldsymbol{B}_{ij}$ is the $i$th row and $j$th column of a matrix $\boldsymbol{B}$. The vector arising from the $j$th column in $\boldsymbol{B}$ is denoted by $\boldsymbol{b}_j$. Positive definiteness of a matrix is denoted by $\succ 0$ and positive semidefiniteness by $\succcurlyeq 0$. The rank of a matrix $\boldsymbol{B}$ is denoted by $\mathcal{R}(\boldsymbol{B})$. We define a matrix $\boldsymbol{B} \in \mathbb{R}^{p \times p}$ as low-rank if $\mathcal{R}(\boldsymbol{B}) \ll p$. We denote the upper triangle of a square symmetric matrix by $\mathcal{U}(\boldsymbol{B})$, with the corresponding vector $\boldsymbol{b}_u$. $\mathcal{P}_\Omega$ is a projection operator that selects the indices in $\Omega$, such that

$$[\mathcal{P}_\Omega(\boldsymbol{B})]_{ij} = \begin{cases} \boldsymbol{B}_{ij} & \text{if } (i, j) \in \Omega \\ \emptyset & \text{if } (i, j) \in \Omega^c \end{cases}$$

We denote the identity matrix by $\boldsymbol{I}$, where entries on the main diagonal are $1$ and all other entries are $0$. We denote the singular value decomposition (SVD) of the matrix $\boldsymbol{B}$ by $\boldsymbol{B} = \boldsymbol{U}\boldsymbol{\Lambda}\boldsymbol{V}^T$, where the $i$th value on the diagonal of $\boldsymbol{\Lambda}$ is denoted by $\lambda_i$. We denote a truncated SVD of rank $r$ by $\boldsymbol{B}_r = SVD_r(\boldsymbol{B}) = \boldsymbol{U}_r\boldsymbol{\Lambda}_r\boldsymbol{V}_r^T$. The norm of a matrix is denoted by $\| \cdot \|$.

## 3.2 Multilayer Graph Structure

We define a multilayer graph using the principles of Kivelä et al. (2014). An $l$-layer multilayer graph is defined as $G_M = (V_M, E_M, V, L)$, where $V$ is the set of vertices or nodes, $L$ is the set of layers, $V_M \subseteq V \times L$ is the set of node-layer tuples, and $E_M \subseteq V_M \times V_M$ defines the edge set describing connections between node-layer tuples. A node, $u$, of a specific layer, $\boldsymbol{\alpha}$, is encoded as $(u, \boldsymbol{\alpha})$. The set of *intralayer* edges, is defined as $E_A = \{((u, \boldsymbol{\alpha}), (v, \boldsymbol{\beta})) \in E_M | \boldsymbol{\alpha} = \boldsymbol{\beta}\}$ while *interlayer* edges are defined as $E_C = \{((u, \boldsymbol{\alpha}), (v, \boldsymbol{\beta})) \in E_M | \boldsymbol{\alpha} \neq \boldsymbol{\beta}\}$ where $E_C = E_M \setminus E_A$.

A supra-adjacency matrix $\boldsymbol{A}$ is a two-dimensional representation of $G_M$ obtained through a mapping process called "flattening" or "matricization" (Kivelä et al., 2014). For multiSLICE, detailed in Section 3.3, we consider a layer-disjoint multilayer network, where each node occupies only one layer. To reduce notation clutter, when $\boldsymbol{\alpha} = \boldsymbol{\beta}$, we simplify the subscript to $\boldsymbol{\alpha}$. Hence, an intralayer adjacency matrix is defined as $\boldsymbol{A}_{\boldsymbol{\alpha}}$, whereas an interlayer adjacency matrix is defined as $\boldsymbol{A}_{\boldsymbol{\alpha\beta}} | \boldsymbol{\alpha} \neq \boldsymbol{\beta}$. $\boldsymbol{A}_{\boldsymbol{\alpha}, \setminus \boldsymbol{\alpha}}$ defines the submatrix formed by the intersection of all rows of $\boldsymbol{\alpha}$ and all columns not in $\boldsymbol{\alpha}$. In general, we use bold subscripts to denote a layer-specific quantity.

**Remark 1.** *For a multilayer graph $G_M$ with a symmetric supra-adjacency matrix $\boldsymbol{A} = \boldsymbol{A}^T$, we make the following connections to the projection operator $\mathcal{P}_\Omega$.*

- *For layer $\boldsymbol{\alpha}$, the intralayer edges $E_{\boldsymbol{\alpha}}$ are encoded by projection of that layer, $\mathcal{P}_{\Omega_{\boldsymbol{\alpha}}}(\boldsymbol{A})$, since we have that $E_{\boldsymbol{\alpha}} = \{((u, \boldsymbol{\alpha}), (v, \boldsymbol{\alpha})) \in E_M \mid u, v \in \omega_{\boldsymbol{\alpha}}\}$.*

- *For $l$ layers, $\Omega$ is the union of individual layers' observed indices, $\Omega = \bigcup_{\boldsymbol{\alpha}=1}^{l}(\Omega_{\boldsymbol{\alpha}}) = \Omega_1 \cup \Omega_2 \cup, \ldots, \cup \Omega_l$. The intralayer edges $E_A$ across all layers are encoded by $\mathcal{P}_\Omega(\boldsymbol{A})$.*

- *The interlayer edges, $E_C = \{((u, \boldsymbol{\alpha}), (v, \boldsymbol{\beta})) \in E_M \mid u \in \omega_{\boldsymbol{\alpha}} \text{ and } v \in \omega_{\boldsymbol{\beta}}, \boldsymbol{\alpha} \neq \boldsymbol{\beta}\}$ are obtained through the complement of $\mathcal{P}_\Omega$, denoted by $\mathcal{P}_{\Omega^c}$. $\mathcal{P}_{\Omega^c}(\boldsymbol{A})$ corresponds to the hidden or unobserved edges of $\boldsymbol{A}$.*

*The problem of multilayer graph estimation can then be seen as a special case of matrix completion in which $\boldsymbol{A}$ is observed in blocks along the main diagonal. More formally, we observe $\mathcal{P}_\Omega(\boldsymbol{A})$ and we wish to recover the full adjacency matrix $\boldsymbol{A}$ by learning the mapping $\mathcal{P}_\Omega^{-1}$.*

To provide some insight, Figure 1 shows a toy example of a 3-layer system with its associated adjacency matrix. In our multilayer system, we are interested in recovering a *weighted* supra-adjacency matrix, in which intralayer edges are organized along the main diagonal in a block-like structure and interlayer edges are in the remaining off-diagonal elements.

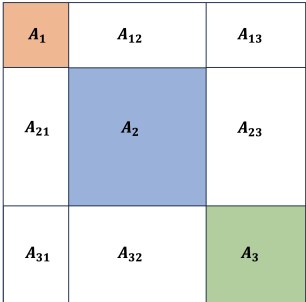 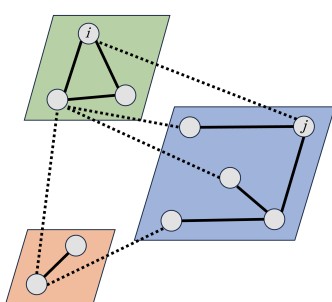

Figure 1: A graphic of a weighted adjacency matrix (left), and the associated multilayer graph (right). Intralayer edges are shown in the graph as solid black lines and color-coded along the main diagonal in a block-like structure ($A_1, A_2, A_3$ in the supra-adjacency matrix). Interlayer edges are encoded with dashed lines in the graph (right) and in the matrix (left) as the remaining off-diagonal blocks (uncolored). Note that this matrix is symmetric, which enforces an undirected structure and ensures that interlayer pairs are transposes of each other ($A_{12} = A_{21}{}^T, A_{13} = A_{31}{}^T, A_{23} = A_{32}{}^T$). The edge between nodes $i, j \in V$ in the graph (right) is expressed as $\{(i, 1), (j, 2)\} \subset E_M$.

### 3.3 Problem Setting

Assume that $\boldsymbol{L}^* \in \mathbb{R}^{p \times p}$ is a weighted supra-adjacency matrix that is composed of square blocks along the main diagonal, corresponding to the observed variables. For matrix $\boldsymbol{L}^*$, we observe structures related only to intralayer edges and denote the projection into the observed space by $\mathcal{P}_\Omega$. $\boldsymbol{L}^*$ and its projection $\mathcal{P}_\Omega(\boldsymbol{L}^*)$ are defined as follows:

$$
\boldsymbol{L}^* = \begin{bmatrix} \boldsymbol{L}_1^* & \boldsymbol{L}_{12}^* & \cdots & \boldsymbol{L}_{1l}^* \\ \boldsymbol{L}_{21}^* & \boldsymbol{L}_2^* & \cdots & \boldsymbol{L}_{2l}^* \\ \vdots & \vdots & \ddots & \vdots \\ \boldsymbol{L}_{l1}^* & \boldsymbol{L}_{l2}^* & \cdots & \boldsymbol{L}_l^* \end{bmatrix}, \quad \mathcal{P}_\Omega(\boldsymbol{L}^*) = \begin{bmatrix} \boldsymbol{L}_1^* & \emptyset & \cdots & \emptyset \\ \emptyset & \boldsymbol{L}_2^* & \cdots & \emptyset \\ \vdots & \vdots & \ddots & \vdots \\ \emptyset & \emptyset & \cdots & \boldsymbol{L}_l^* \end{bmatrix}.
$$

Note that the low-rank latent component corresponding to layer $\alpha$ is given by $\boldsymbol{L}_\alpha^* = \mathcal{P}_{\Omega_\alpha}(\boldsymbol{L}^*)$ for $\alpha \in \{1, \cdots, l\}$. There are also $l$ sparse intralayer matrices, the set of which is denoted by $\{\boldsymbol{S}^*\}_{\alpha=1}^l$.

We assume that the inverse covariance matrix $\boldsymbol{\Sigma}_\alpha^{*-1}$ originates from the sum of a sparse component $\boldsymbol{S}_\alpha{}^*$ and a low-rank (or latent, used interchangeably) component $\boldsymbol{L}_\alpha{}^*$ such that $\boldsymbol{\Sigma}_\alpha^{*-1} = \boldsymbol{S}_\alpha^* + \boldsymbol{L}_\alpha^*$, where $\boldsymbol{L}_\alpha^* = \mathcal{P}_{\Omega_\alpha}(\boldsymbol{L}^*)$. We also assume $\boldsymbol{X}_1 \ldots \boldsymbol{X}_l \in \mathbb{R}^{n_\alpha \times p_\alpha}$ are i.i.d. draws from a multivariate normal distribution, $\boldsymbol{X}_\alpha \sim \mathcal{N}(\mu_\alpha, \boldsymbol{\Sigma}_\alpha^*)$. For the population covariance matrix $\boldsymbol{\Sigma}_\alpha^* \succ 0$ and population mean $\mu_\alpha$, the finite sample realization is the sample covariance matrix,

$$
\tilde{\boldsymbol{\Sigma}}_\alpha = \frac{1}{n_\alpha - 1} \sum_{i=1}^{n_\alpha} (\boldsymbol{x}_{\alpha i} - \bar{\boldsymbol{x}}_\alpha)(\boldsymbol{x}_{\alpha i} - \bar{\boldsymbol{x}}_\alpha)^T,
$$

where $\boldsymbol{x}_{\alpha i}$ and $\bar{\boldsymbol{x}}_\alpha$ are $p_\alpha$ dimensional vectors of the $i$th sample and sample means, respectively, from the $\alpha$th layer. Ignoring the $\mu_\alpha$ term, the log-likelihood function is given by

$$
\mathcal{L}(\boldsymbol{S}_\alpha + \boldsymbol{L}_\alpha; \tilde{\boldsymbol{\Sigma}}_\alpha) = \log \det(\boldsymbol{S}_\alpha + \boldsymbol{L}_\alpha) - \operatorname{tr}(\tilde{\boldsymbol{\Sigma}}_\alpha(\boldsymbol{S}_\alpha + \boldsymbol{L}_\alpha)).
$$

The joint multilayer likelihood function, then, is given by the sum of individual layerwise likelihoods

$$
\mathcal{L}_M(\{\boldsymbol{S}\}_{\alpha=1}^l, \boldsymbol{L}; \{\tilde{\boldsymbol{\Sigma}}\}_{\alpha=1}^l) = \sum_{\alpha=1}^l \mathcal{L}(\boldsymbol{S}_\alpha, \mathcal{P}_{\Omega_\alpha}(\boldsymbol{L}); \tilde{\boldsymbol{\Sigma}}_\alpha). \tag{1}
$$

**Remark 2.** *By imposing a joint latent matrix $\boldsymbol{L}^*$, the number of nodes ($p_\alpha$) between layers can vary. By focusing on the inverse covariance (precision) matrix rather than the layerwise data vectors (which may be different lengths depending on the number of samples), we do not impose the constraint that the number of samples ($n_\alpha$) be identical across layers.*

In short, we model a multilayer network as a product of layer-specific Gaussian densities whose precision matrices share a joint parameterization in the latent space. We set this up by decomposing the

precision matrix of each modality into a sparse component unique to that modality and a shared latent component that captures both the within-modality and the cross-modality edges of the multilayer network. Our objective is to maximize (1) by estimating both the sparse layerwise components as well as the low-rank joint latent component. In Figure 2, we show both the forward data-generating process and the proposed reverse estimation method. The data-generating process comprises two steps: the first step is a projection of the latent space into the observed space via $\mathcal{P}$, and the second step is mixing with sparse components to generate the precision matrix. This two-step decomposition is important in the next section, where we propose estimating parameters via the reverse process.

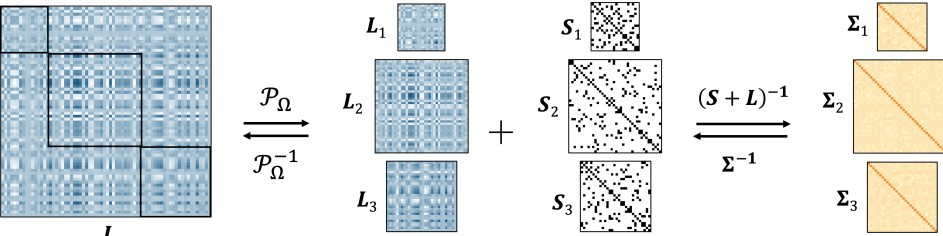

Figure 2: An illustration of the data-generating (forward) and parameter estimation (reverse) processes for a three-layer system. $\boldsymbol{L}$ denotes the joint latent space parameterized by a weighted supra-adjacency matrix, $\mathcal{P}_\Omega$ denotes the projection into observed layers, and its inverse is $\mathcal{P}_\Omega^{-1}$. The observed $\boldsymbol{S}$ and $\boldsymbol{L}$ correspond to edge weights from the intralayer components.

## 4   Methodology

In (1), the summation over $l$ layers decomposes the objective into independent latent variable GGMs. To this end, we introduce the underline{multi}layer underline{s}parse + underline{l}ow-rank underline{i}nverse underline{c}ovariance underline{e}stimator (multiSLICE), which minimizes the following function,

$$\underbrace{\sum_{\substack{\boldsymbol{\alpha}=1 \\ \underbrace{}_{\text{layers}}}}^{l}}(\underbrace{-\mathcal{L}(\hat{\boldsymbol{S}}_{\boldsymbol{\alpha}};(\tilde{\boldsymbol{\Sigma}}_{\boldsymbol{\alpha}}^{-1}-\hat{\boldsymbol{L}}_{\boldsymbol{\alpha}})^{-1})+\rho\|\hat{\boldsymbol{S}}_{\boldsymbol{\alpha}}\|_1}_{\text{penalized negative log likelihood}}+\underbrace{\|\tilde{\boldsymbol{\Sigma}}_{\boldsymbol{\alpha}}(\hat{\boldsymbol{S}}_{\boldsymbol{\alpha}}+\hat{\boldsymbol{L}}_{\boldsymbol{\alpha}})-\boldsymbol{I}\|_F^2}_{\text{covariance fidelity}}) \tag{2}$$

$$\text{s.t. } \mathcal{R}(\hat{\boldsymbol{L}})=r, \text{ where } 0<r<p\,.$$

Here, $\rho$ is a tuning parameter for the sparsity in $\hat{\boldsymbol{S}}$ and $r$ is the pre-specified rank of the latent matrix $\hat{\boldsymbol{L}}$. The objective function is decomposed into a penalized negative log-likelihood term and a covariance fidelity term. We propose a two-stage algorithm to solve (2), which follows Figure 2. In the first stage, individual latent variable GGMs are estimated, and in the second stage, matrix completion is applied to the latent estimates using the block singular value decomposition algorithm (Bishop and Yu, 2014). Algorithm 1 describes an alternating descent algorithm which we refer to as sparse + low-rank inverse covariance estimation (SLICE). Notice that it can also be used with various other penalties on $\hat{\boldsymbol{S}}$ (e.g., SCAD; Fan et al. 2009) or estimators (e.g., CLIME; Cai et al. 2011) by simply substituting in a different estimator for $\hat{\boldsymbol{S}}$ (see the Supplementary Materials for more details).

---

**Algorithm 1** Sparse + low-rank inverse covariance estimation (SLICE) with GLASSO

**Inputs:** $\tilde{\boldsymbol{\Sigma}}, \rho, r, maxiter, tol$
$\hat{\boldsymbol{L}}^0 = \hat{\boldsymbol{S}}^0 = 0$
   **for** $i = 1$ to $maxiter$ **do**
      $\hat{\boldsymbol{L}}^{(i)} := SVD_r(\tilde{\boldsymbol{\Sigma}}^{-1} - \hat{\boldsymbol{S}}^{(i-1)})$
      $\hat{\boldsymbol{S}}^{(i)} := \text{GLASSO}((\tilde{\boldsymbol{\Sigma}}^{-1} - \hat{\boldsymbol{L}}^{(i)})^{-1}, \rho)$
   **end for** $\Delta\hat{\boldsymbol{L}}$ AND $\Delta\hat{\boldsymbol{S}} < tol$ OR $\Delta\mathcal{L}(\hat{\boldsymbol{S}} + \hat{\boldsymbol{L}}) < tol$
**Outputs:** $\hat{\boldsymbol{S}}, \hat{\boldsymbol{L}}$

---

**Algorithm 2** Multilayer SLICE (multiSLICE) estimation

**Inputs:** $\{\hat{\boldsymbol{\Sigma}}\}_{\boldsymbol{\alpha}=1}^l, \rho, r, maxiter, tol$
Apply SLICE on $\hat{\boldsymbol{\Sigma}}_{\boldsymbol{\alpha}}$ for all $\boldsymbol{\alpha}$
Initialize $\boldsymbol{H}^{p \times r} = 0$
   **for** $k = 1$ to $l$ **do**
      $\boldsymbol{U}_r\boldsymbol{\Lambda}_r\boldsymbol{V}_r^T := SVD_r(\hat{\boldsymbol{L}}_{\boldsymbol{\alpha}})$
      $\boldsymbol{H}_{\boldsymbol{\alpha}} := \boldsymbol{U}_r\boldsymbol{\Lambda}_r^{1/2}$
   **end for**
$\hat{\boldsymbol{L}} := \boldsymbol{H} \times \boldsymbol{H}^T$
**Outputs:** $\{\hat{\boldsymbol{S}}\}_{i=1}^l, \hat{\boldsymbol{L}}$

---

We apply SLICE to multilayer networks through a block-coordinate approach, using it independently for each layer. The estimated $\hat{L}_\alpha$ for each layer is then used for the matrix completion step in the second part of the multiSLICE algorithm. The complete multiSLICE method is detailed in Algorithm 2. For the matrix completion portion, Algorithm 2 first computes per-layer SVDs, $L_i = U_i \Lambda_i U_i^T$, $i = 1, \ldots, l$, then intersects each pair of latent subspaces by forming $L_{ij} = U_i \Lambda_i^{1/2} \Lambda_j^{1/2} U_j^T$, and finally assembles all blocks $\{L_{ij}\}$ in the global low-rank matrix, $L$. As an illustrative example, we can consider a two-layer case, where we reconstruct the interlayer latent adjacency matrix $L_{12} = U_1 \Lambda_1^{1/2} \Lambda_2^{1/2} U_2^T$ before assembling them into $L$.

Given assumptions for exact recovery in Theorem 5.1, the rank of each submatrix $L_\alpha^*$ must equal that of the overall $L^*$ across all $\alpha$. From Remark 2, we know that depending on $p_\alpha$ and $n_\alpha$ in each layer, different values of $\rho$ can be specified for each independent SLICE model. To select $\rho$ and $r$, we suggest a $k$-fold cross-validation over a grid, where the combined values are based on log-likelihood. Simulation experiments elucidating the sensitivity of multiSLICE to these choices are provided in the Supplementary Materials.

## 5 Theoretical analysis

We establish consistency conditions for the sparse components and the low-rank joint latent component for our multiSLICE approach. Our proof considers each independent SLICE and matrix completion step separately. Our objective is to show the conditions under which we can exactly recover $L^*$ as the sample size in each layer, $n_\alpha \to \infty$. To recover the full $L^*$, we must *exactly* recover $L^*$ from only the observed entries denoted by $\mathcal{P}_\Omega(L^*)$. To do so, we use a result from Liu et al. (2017), that describe the general conditions under which exact matrix completion is possible. More formally, Liu et al. (2017) describe the conditions where $\mathcal{P}_\Omega^{-1}$ exists, which leads us to our first result in Theorem 5.1.

**Theorem 5.1** (Recovery of $L^*$). *Let $L^* \succcurlyeq 0$ with rows and columns indexed by $\omega = \{1, \ldots, p\}$. Let $\Omega = \cup_{\alpha=1}^l \Omega_\alpha$, where $\Omega_\alpha = \omega_\alpha \times \omega_\alpha$ and $\omega_\alpha \subseteq \omega$. If $\mathcal{R}(L_\alpha^*) = \mathcal{R}(L^*) \ \forall \alpha = 1, \ldots, l$ and $|\cup_{\alpha=1}^l \omega_\alpha| \geq |\omega|$, then we have that $\mathcal{P}_\Omega(L^*)$ is invertible.*

This result indicates that for exact recovery we only require that each submatrix formed by $\mathcal{P}_{\Omega_\alpha}(L^*)$ be of the same rank as $L^*$. Next, we require standard assumptions for sub-Gaussianity, appropriate regularization, and bounds on the eigenvalues of $S_\alpha^*$ and $L_\alpha^*$ to recover $S_\alpha^*$ and $L_\alpha^*$ (Theorem 5.2).

**Remark 3.** *By Lemma 6.8 of Liu et al. (2017), the $\Omega/\Omega^T$-isomeric condition is equivalent to requiring the operators $\mathcal{P}_{U^*}\mathcal{P}_\Omega\mathcal{P}_{U^*}$ and $\mathcal{P}_{V^*}\mathcal{P}_\Omega\mathcal{P}_{V^*}$ be invertible. Hence, these invertibility conditions are exactly the identifiability requirements for the parameter matrix $L^*$.*

**Theorem 5.2** (multiSLICE joint $\{\hat{S}\}_{\alpha=1}^l$ and $\hat{L}$ consistency). *Let $L^* \succcurlyeq 0$ with rows and columns indexed by $\omega = \{1, \ldots, p\}$. Let $\Omega = \cup_{\alpha=1}^l \Omega_\alpha$, where $\Omega_\alpha = \omega_\alpha \times \omega_\alpha$ and $\omega_\alpha \subseteq \omega$. Additionally, let $\{S^*\}_{\alpha=1}^l$ be the set of true sparse matrices. Then, we have $\hat{S}_{\alpha off} \overset{P}{\to} S_{\alpha off}^*$ and $\hat{L} \overset{P}{\to} L^*$ as $n_\alpha \to \infty$, $\forall \alpha = 1 \ldots l$.*

Given the independence of SLICE models, we show the model selection consistency of the sparse component in a similar manner. We require standard assumptions for the minimum signal strength of $S^*$ and irrepresentability. This leads us to the result of Theorem 5.3.

**Theorem 5.3** (Model Selection Consistency of $\hat{S}_\alpha$). *For the multiSLICE estimator with L1 regularization for $\hat{S}$ in layer $\alpha$, we have*

$$\mathbb{P}\left(sign(\hat{S}_{\alpha ij}^\lambda) = sign(S_{\alpha ij}^*), \forall i, j \in \hat{S}_\alpha^\lambda\right) \geq 1 - \frac{1}{p^{\tau-2}} \to 1.$$

There are further considerations that arise when $p$ and/or $n$ vary between layers.

**Remark 4.** *We apply SLICE models independently, and the conditions for model selection consistency in each layer for the value of $\rho$ may be different if the number of observed variables in the layer $p$ and/or the sample size $n$ is different. In such cases, the bounds for $\|\hat{S}_{\alpha off} - S_{\alpha off}^*\|_F$, $\|\hat{L}_\alpha - L_\alpha^*\|_\infty$ are also different. Layers with lower $p_\alpha$ and higher $n_\alpha$ therefore require less regularization (through $\rho$) and have tighter bounds on $\hat{S}$ and $\hat{L}$.*

Theorem 5.2 establishes the consistency of multiSLICE. However, to establish a rate, we require a further assumption. In particular, we assume that $\|[\hat{L} - L^*]_{\backslash \alpha, \alpha}\|_\infty = \mathcal{O}(\|\hat{L}_\alpha - L_\alpha^*\|)$, where for some layer $\alpha$, the rate of the unobserved portions is the same as the rate of the observed portions, up to a constant. With this, we can prove the result in Theorem 5.4.

**Theorem 5.4** (Rate for joint low-rank, latent space). *Let $L^* \succcurlyeq 0$ with rows and columns indexed by $\omega = \{1, \ldots, p\}$. Let $\Omega = \cup_{\alpha=1}^l \Omega_\alpha$, where $\Omega_\alpha = \omega_\alpha \times \omega_\alpha$ and $\omega_\alpha \subseteq \omega$. If $\|[\hat{L} - L^*]_{\backslash \alpha, \alpha}\|_\infty = \mathcal{O}(\|\hat{L}_\alpha - L_\alpha^*\|_\infty)$, then we have that*

$$\|\hat{L} - L^*\|_\infty \lesssim \sum_{\alpha=1}^l C \sqrt{\frac{\log p_\alpha}{n_\alpha}} \quad \text{for some constant } C.$$

**Remark 5.** *From Theorem 5.4, we have that the $\infty$-norm of the joint space is bounded by the sum of the layerwise $L_\infty$ norms. For a two-layer system, the following is implied: The overall bounds on $\hat{L}$ improve as we increase the sample size from one domain, $\alpha$ or $\beta$, even while holding the other number of samples constant. However, we can only recover $L^*$ exactly when we have exactly recovered $L_\alpha^*$ and $L_\beta^*$.*

This two-layer intuition can be extended to a network of any arbitrary size by considering any pair of layers $\alpha, \beta \in 1, \ldots, l$. These phenomena are made evident in the simulation studies in Section 6.

# 6 Simulated data study

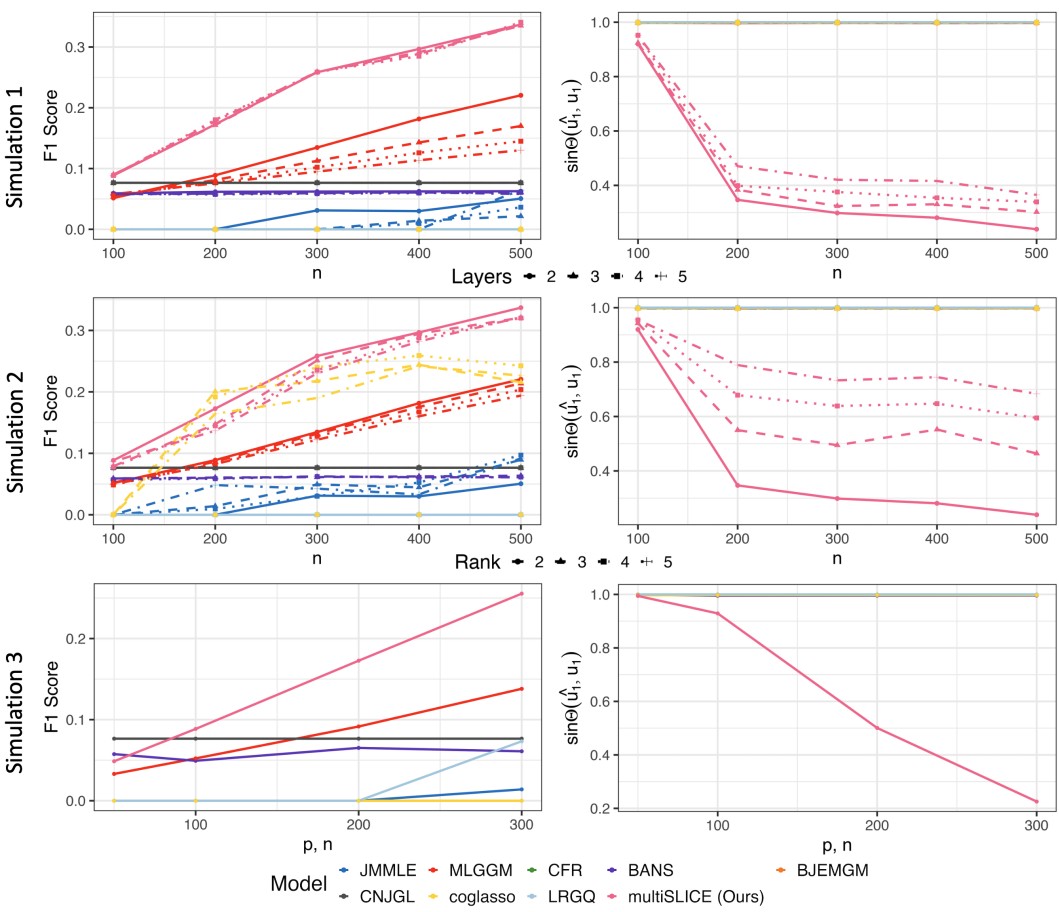

Figure 3: The effects of changing $l$, the number of layers (simulation 1) and $\mathcal{R}(L^*)$, the rank of the latent parameter matrix (simulation 2), and scaling $p$ and $n$ at the same rate (simulation 3). Each simulation has 100 iterations.

We consider two challenging simulation studies, which are inspired by structures in neuroimaging (Yatsenko et al., 2015). For each simulation, we generate $\boldsymbol{S}_{\boldsymbol{\alpha}}^*$ and $\boldsymbol{L}^*$. To generate the sparse component for the $\boldsymbol{\alpha}$th layer, $\boldsymbol{S}_{\boldsymbol{\alpha}}^*$, we first use an initial value for the main diagonal, denoted by $\beta$ and a decay rate, denoted by $\zeta$. The $ij$th element is defined as

$$
S_{\boldsymbol{\alpha} ij}^* = \begin{cases} \beta e^{\zeta|i-j|} & \text{if } \beta e^{\zeta|i-j|} \geq \psi, \\ 0 & \text{if } \beta e^{\zeta|i-j|} < \psi. \end{cases}
$$

We then permute $\boldsymbol{S}_{\boldsymbol{\alpha}}^*$ over rows and columns to randomize the structure. We set $\beta = 1.5, \zeta = 2, \psi = 0.01$, and $p_{\boldsymbol{\alpha}} = 100$ for all simulations. To generate $\boldsymbol{L}^*$, we construct a $p \times r$ binary matrix $Z$ by, for each row $i$, selecting one of the $r$ columns uniformly at random and setting $Z_{ij} = 1$, with all other entries in row $i$ equal to zero. Then, we obtain $\boldsymbol{L}^* \leftarrow \beta \times \boldsymbol{Z}\boldsymbol{Z}^T$. For simulation 1, we fix $\mathcal{R}(\boldsymbol{L}^*) = 2$ and vary $l$; for simulation 2, we fix $l = 2$ and vary $\mathcal{R}(\boldsymbol{L}^*)$; for simulation 3, we fix $\mathcal{R}(\boldsymbol{L}^*) = 2$ and $l = 2$, and vary $p$ and $n$ jointly. We evaluate estimates of $\hat{\boldsymbol{S}}$ using the F1 score for sparsity, which is typical in the sparse GGM literature (Wang and Allen, 2023). We evaluate $\hat{\boldsymbol{L}}$ using the angle between the first eigenvector $\hat{\boldsymbol{L}}$ and $\boldsymbol{L}^*$, denoted by $\sin\theta(\hat{\boldsymbol{u}}_1, \boldsymbol{u}_1^*)$, which is a natural choice for comparing the estimated and true low-rank parameters (Athreya et al., 2018).

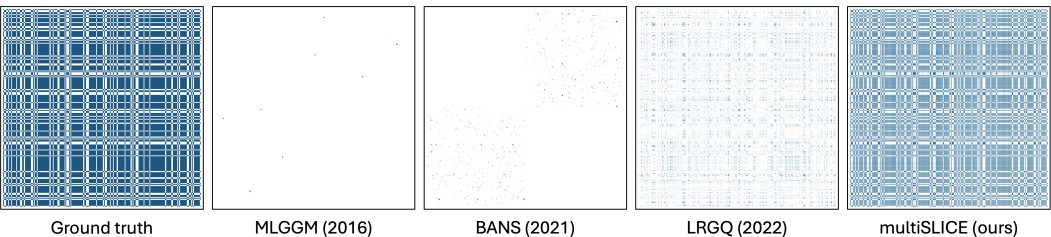

| Ground truth | MLGGM (2016) | BANS (2021) | LRGQ (2022) | multiSLICE (ours) |

Figure 4: Recovery of $\boldsymbol{L}^*$ with $l = 2$, $\mathcal{R}(\boldsymbol{L}^*) = 2$, $n = 500$ for methods with non-zero estimates.

Figure 3 shows the results of our simulation study. Across all simulations, we find that multiSLICE outperforms all other methods in the F1 score, indicating improved estimation of $\boldsymbol{S}^*$. In addition, it is the only method that converges toward the true $\boldsymbol{L}^*$, as indicated by the decreasing $\sin\theta(\hat{\boldsymbol{u}}_1, \boldsymbol{u}_1^*)$ over increasing $n$.

Also, for multiSLICE, the number of layers does not appear to greatly affect the F1 score and $\sin\theta(\hat{\boldsymbol{u}}_1, \boldsymbol{u}_1^*)$, while $\mathcal{R}(\boldsymbol{L}^*)$ has a stronger influence. In Simulation 1 and 2, for all methods, increasing $l$ or $\mathcal{R}(\boldsymbol{L}^*)$ has a negative impact on performance. From the related methods, we find that MLGGM has the second-best performance in simulation 1, and coglasso has the second-best performance in simulation 2. There is a vast difference in coglasso's performance between Simulations 1 and 2, suggesting that it is more sensitive to the number of layers than $\mathcal{R}(\boldsymbol{L}^*)$. Figure 4 shows the recovered latent parameter matrices.

Figure 5 shows the results of simulating a two-layer system with $\mathcal{R}(\boldsymbol{L}^*) = 2$, with varying $n_1$ and $n_2$, and applying multiSLICE. Our results suggest that increasing the sample size in one layer improves recovery of the joint latent structure as $n_1$ or $n_2$ increases.

Lastly, we conduct simulation studies following Wainwright (2009) to validate our theoretical work. In all 100 iterations, we simulate a two-layer system, apply multiSLICE, then compute the relevant error norm, and declare recovery successful if that norm falls below a constant, as predicted by Lemma 9.1. In

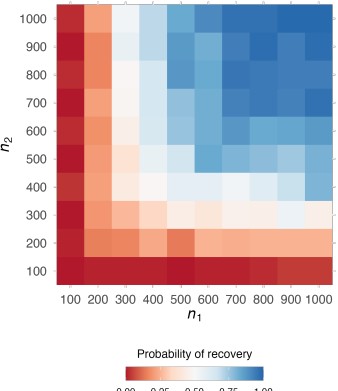

Figure 5: $\boldsymbol{L}$ recovery for different layer sample sizes $n_1$, $n_2$ with constant $l = 2$, $\mathcal{R}(\boldsymbol{L}^*) = 2$.

Figure 6, we plot the empirical success probability against the raw sample size $n_{\boldsymbol{\alpha}}$, and scaled sample size, defined as

$$
n_{\boldsymbol{S}}' = \frac{n_{\boldsymbol{\alpha}}}{s_{\boldsymbol{\alpha}} \log p_{\boldsymbol{\alpha}}}, \qquad n_{\boldsymbol{L}}' = \frac{n_{\boldsymbol{\alpha}}}{C_1 \log p_{\boldsymbol{\alpha}}},
$$

where $s_{\boldsymbol{\alpha}}$ and $p_{\boldsymbol{\alpha}}$ are the number of nonzeros and variables in modality $\boldsymbol{\alpha}$, respectively, and $C_1$ is defined in Lemma 9.1. We find that by scaling the sample sizes by the appropriate factors, the probability of recovery curves collapse toward a common transition point as predicted by our theory. Details of hyperparameter selection in all simulations and additional information are in the Supplementary Materials.

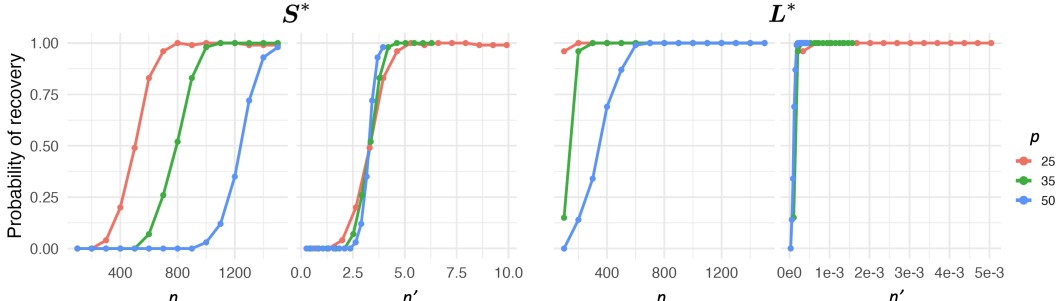

Figure 6: The probability of successful recovery of $\boldsymbol{S}^*$ (left panel) and $\boldsymbol{L}^*$ (right panel). In each panel, the left subplot x-axis uses the sample size $n$, whereas the right x-axis uses the scaled sample sizes for $\boldsymbol{S}^*$ and $\boldsymbol{L}^*$, respectively.

# 7 Multimodal neuroimaging data study

We apply multiSLICE and competitor methods to a multimodal neuroimaging dataset from Wakeman and Henson (2015). In this dataset, 16 subjects are scanned during the presentation of three different facial stimuli. "Famous" and "Unfamiliar" faces are those of people who are publicly well known and those of people who are not, respectively. The "Scrambled" group is a set of images that have the outline or general shape of a face but are filled in with white noise; these images serve as a control stimulus. Each subject has data related to structure (sMRI) and function (MEG, fMRI). Pre-processing steps and further details are in the Supplementary Materials, and all experiments are run on a M1 MacBook Pro with 16GB of RAM using R 4.4.3.

Each modality $\boldsymbol{\alpha}$ has different $p_{\boldsymbol{\alpha}}$ and $n_{\boldsymbol{\alpha}}$, which is typical in neuroimaging. We exclude methods that assume an equal sample size $n_{\boldsymbol{\alpha}}$ between layers (MLGGM, BANS, JMMLE, coglasso), as this assumption is violated in our dataset. Due to excessive run-time, CFR was omitted. For methods requiring the same $p_{\boldsymbol{\alpha}}$ across modalities (CNJGL, BJEMGM), we apply SVD to project the data into the smallest joint subspace, corresponding to MEG ($p_{\text{MEG}} = 52$). We then apply CNJGL, BJEMGM, LRGQ, multiSLICE to estimate networks independently for each subject and stimulus.

| | Famous | | Unfamiliar | | Scrambled | |
|---|---|---|---|---|---|---|
| | $Q(\hat{\boldsymbol{S}})$ | $H(\hat{\boldsymbol{L}})$ | $Q(\hat{\boldsymbol{S}})$ | $H(\hat{\boldsymbol{L}})$ | $Q(\hat{\boldsymbol{S}})$ | $H(\hat{\boldsymbol{L}})$ |
| CNJGL (2014) | 0.107 (9.65e-03) | N/A | 0.106 (1.01e-02) | N/A | 0.104 (1.08e-02) | N/A |
| BJEMGM (2019) | 0.084 (1.25e-02) | N/A | 0.084 (9.09e-03) | N/A | 0.079 (1.27e-02) | N/A |
| LRGQ (2022) | 0.112 (1.21e-01) | 1.29 (6.63e-02) | 0.122 (1.18e-01) | 1.29 (6.38e-02) | 0.130 (1.07e-01) | 1.28 (7.09e-02) |
| multiSLICE (ours) | **0.170 (1.86e-03)** | **0.626 (4.53e-02)** | **0.171 (1.64e-03)** | **0.631 (4.02e-02)** | **0.170 (1.28e-03)** | **0.660 (1.17e-01)** |

Table 2: The estimated modularity of the sparse intralayer graphs, $Q(\hat{\boldsymbol{S}})$, and the multilayer von Neumann entropy for the latent supra adjacency matrix, $H(\hat{\boldsymbol{L}})$. Bold values are the best results across methods, highest for $Q(\hat{\boldsymbol{S}})$ and lowest for $H(\hat{\boldsymbol{L}})$, for each visual stimulus (Famous, Unfamiliar, and Scrambled). Entries show the mean (standard deviation) across the 16 subjects.

To compare estimates of $\hat{\boldsymbol{S}}$ from each method, we use modularity (Bullmore and Sporns, 2009), which measures how well communities are separated within the intralayer graphs. This is calculated using $Q(\hat{\boldsymbol{S}}) = \frac{1}{2m} \operatorname{tr}(\boldsymbol{C}^T \boldsymbol{B} \boldsymbol{C})$, where $\boldsymbol{B} = \hat{\boldsymbol{S}} - \frac{\mathbf{k}\mathbf{k}^T}{2m}$, where $\boldsymbol{C}$ is a matrix of community assignments, $m$ is the total number of edges and $\mathbf{k}$ is a vector of node degrees. To compare estimates of $\hat{\boldsymbol{L}}$ from each method, we use the multilayer von Neumann entropy, $H$, (De Domenico et al., 2015), where lower values are preferred, indicating higher order in the network. The von Neumann entropy is

defined as $H(\hat{L}) = -\sum_{i=1}^{p} \lambda_{\phi_i} \ln(\lambda_{\phi_i})$, where $\phi$ is $\hat{L}$, normalized to a trace of 1, $\phi = \frac{\hat{L}}{\text{tr}(\hat{L})}$. Table 2 shows the results. Across all measures and face stimuli, multiSLICE has the best performance, supporting the efficacy of our method. LRGQ has the second-best performance across all measures, followed by CNJGL and BJEMGM, suggesting that the multiSLICE model is more realistic and yields more favorable estimates of both $\hat{S}$ and $\hat{L}$. Figure 7 shows the multiSLICE adjacency estimates for the different face stimulus conditions. Although they all share a low-rank structure, capturing the common task-related variance, each stimulus produces a unique pattern that highlights stimulus-specific multilayer networks.

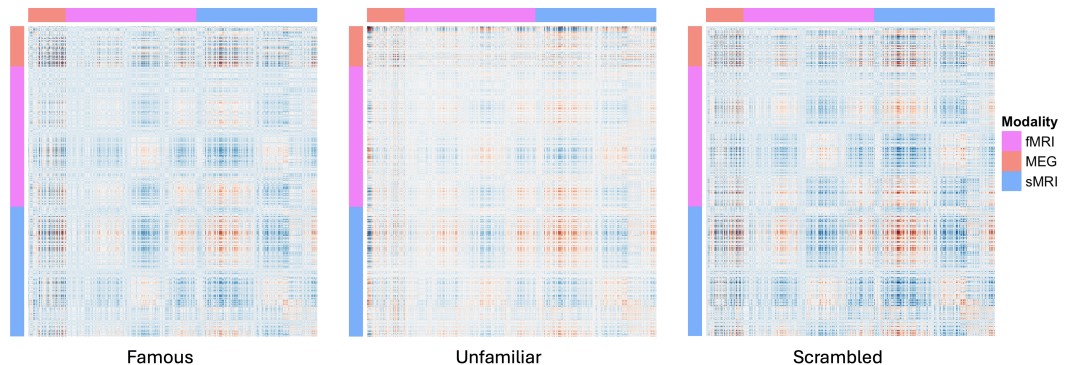

Figure 7: The estimated supra adjacency matrices, $\hat{L}$, obtained from multiSLICE for each visual stimuli presented to subject 1 of the Wakeman and Henson (2015) dataset.

# 8 Discussion

We highlight three key regimes in which our model's guarantees may not hold in practice. The first is that if measurements from different modalities are gathered simultaneously, our assumption of independent modality measurements is violated. Second, any form of temporal or spatial correlation across observations breaches the i.i.d. requirement underpinning our per-modality covariance estimators. Lastly, while our sparse, joint-latent parameterization performs well for neuroimaging data, it remains untested in other application areas. As such, practitioners should validate its suitability before applying it to novel domains.

Future studies could adapt multiSLICE to capture sample-to-sample dependencies via matrix-variate extensions that regularize the row precision matrix in a joint manner. Time-varying and longitudinal domains are also noteworthy, as it is well known that brain imaging data can be non-stationary (Fox et al., 2005; Eichele et al., 2008; Doucet et al., 2012). For example, for the time-varying case, one could formulate the objective function to jointly estimate $\{L^{(t)}\}_{t=1}^{T}$, with penalties enforcing smoothness or sparsity in time as in Hallac et al. (2017). For strictly repeated (longitudinal) measurements, one only needs to treat each time point as a separate "layer" in multiSLICE, ordering covariance inputs by time so that the interlayer estimates $\{L_{t,t+1}\}$ capture the within-subject evolution directly.

We also consider extending multiSLICE into end-to-end deep-learning workflows by using its outputs, namely the sparse $S$ and low-rank $L$ estimates to define graph inputs for GNNs. For example, Do et al. (2023) showed that decoupling graph construction (via Graphical LASSO) from GNN training both accelerates convergence and improves performance, while Sriramulu et al. (2023)'s adaptive dependency learning graph neural networks (ADLNN) uses Graphical LASSO as a structural prior before refining edges within a GNN. In neuroimaging, Wang et al. (2021); Yu et al. (2022); Thapaliya et al. (2025) utilize graphs representations of regional brain activity for downstream prediction tasks. These single-modality methods underscore how graphical models yield effective, interpretable graphs from multivariate data; multiSLICE generalizes this to a multimodal setting, filling a clear gap.

The iterative steps of Algorithm 2 can be unrolled into a module and embedded directly into differentiable architectures. Alternatively, one can amortize the mapping from sample covariances to $(S, L)$ via a small neural network, similar to Belilovsky et al. (2017), and feed its output into downstream GNNs. In the case of non-Gaussian data, such as raw images, audio, or video, multiSLICE could be used downstream of modality-specific feature extractors (e.g. a CNN).

## Acknowledgements

We would like to sincerely thank the reviewers for their helpful feedback and comments throughout the submission, which have led to several improvements of the paper. The first author was supported by the H. Jean McDiarmaid scholarship provided by the Faculty of Medicine and Dentistry, University of Alberta. The second author was supported by the Natural Sciences and Engineering Research Council of Canada (NSERC) Discovery Grant RGPIN-2024-06102. The third author was partially supported by the National Science Foundation (NSF) Grant DMS-2324489.

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
