# OpenReview forum: "A Latent Multilayer Graphical Model For Complex, Interdependent Systems"
_NeurIPS.cc/2025/Conference — NeurIPS 2025 poster_

### Official Review · Reviewer_pLq3 · 2025-06-20

**Clarity:** 3
**Significance:** 2
**Originality:** 2
**Rating:** 4
**Confidence:** 3

**Summary:**

The paper addresses the problem of recovering dependencies among individual systems in multimodal data by modeling them as a multilayer graph, where each system is represented by a latent-variable Gaussian graphical model (GGM). The primary contribution is a statistical model that estimates interlayer edges across disjoint GGMs that vary in structure, number of nodes, and sample sizes.

The authors provide theoretical guarantees for the recoverability of the joint space, exploring how interlayer interactions impact joint parameter estimation. They also derive theoretical bounds on these relationships. To the best of their knowledge, this is the first model capable of estimating interlayer connections in a multilayer network that handles heterogeneous dimensions, sample sizes, and scales to multiple modalities.

**Questions:**

N/A

**Ethical Concerns:**

["NO or VERY MINOR ethics concerns only"]

**Final Justification:**

Following my discussion with the authors, I decide to improve the rating to Borderline Accept given their additional discussions on the applicability of the proposed method in deep learning frameworks. The borderline evaluation pertains to the limit of the current discussion to high-level ideas, rather than a detailed technical discussion that should have been included in the submitted version. Overall, the paper itself is a sound statistical work itself. Although the contribution to me remains an incremental contribution to the statistics community, I think the problem setup and the completeness of the results, especially regarding the recovery of latent factors, can somewhat inspire the design of new learning algorithms.

**Limitations:**

Yes. The limitations have been discussed.

**Quality:**

3

**Strengths And Weaknesses:**

### **Strengths:**

The paper is well written and well motivated, with a clear problem formulation. The proposed framework is flexible and shows superior empirical performance compared to baseline methods with insightful ablation studies. It includes theoretical recovery guarantees for the joint parameters. The approach can also handle heterogeneous systems and is scalable to multiple modalities.

### **Weaknesses:**

The paper makes several assumptions which may limit generalizability in real-world data. These assumptions include:
1. the individual models are disjoint Gaussians
2. known intralayer connectivity and assumes that data is generated from a self-sufficient model with that known structure. From my understanding from Line 138, the data is assumed to be not influenced by the interlayer latent dependencies.
3. prior knowledge of the maximum number of variables (p), including latent ones
4. decomposability of the joint likelihood into layer-wise summations for tractability. The decomposition is made with few justifications. It would have been more useful if the authors could provide more convincing justifications for this design choice if there were any.

While the paper provides a solid solution to the problem at hand, in all honestly, I am unsure whether it fits into the landscape of NeurIPS since the authors solely employ traditional statistical methods and the paper lacks application of neural networks. The authors should enhance the paper's relevancy and impact by demonstrating applications in deep learning contexts to align with contemporary research directions.

---

> ### Author Rebuttal · Authors · 2025-07-31
>
> **Weaknesses**
>
> *The paper makes several assumptions which may limit generalizability in real-world data. These assumptions include: The individual models are disjoint Gaussians with known intralayer connectivity and assumes that data is generated from a self-sufficient model with that known structure. From my understanding from Line 138, the data is assumed to be not influenced by the interlayer latent dependencies.*
>
> Thank you for this comment. We emphasize that in multiSLICE, no layer is treated in isolation — each layer utilizes the same joint low‑rank latent matrix, hence its precision matrix combines that shared block with a layer‑specific sparse component. As a result, the structure and parameters estimated in any one layer are directly influenced by the information from all other layers via the common latent structure.
>
> Although we focus on the Gaussian case for clarity, multiSLICE naturally extends to non‑Gaussian settings. For example, we could use the approach of Liu et al. (2009) and extend our method to a semi-parametric domain. Further extensions to exponential family graphical models are also worth exploring (Yang et al., 2015).. We have added a note about these potential extensions to the discussion.
>
> *Prior knowledge of the maximum number of variables (p), including latent ones*
>
> For multiSLICE, no extra prior knowledge of $p$ or the latent dimension is required beyond the data itself. The total variable count $p$ is simply the sum of measured features across modalities. The latent rank $r$ is treated as a tunable hyperparameter rather than an assumed constant: we select $r$ (and the sparsity penalty $\rho$) via 3‑fold cross‑validation on held‑out likelihood.
>
> In response to your comment, we have adjusted the main article, specifically Sections 6 and 7, to better describe the procedure we use.  This is also implemented in our codebase on the anonymous GitHub repository as the `cv.multislice(Xs, folds, rhos, rs)` function, where `Xs`, `folds`, `rhos`, and `rs` are the input data, number of folds for cross-validation, vector of candidate $\rho$ values, and vector of candidate $r$ values, respectively. A set of simulation studies looking at the effects of $\rho$ and $r$ on parameter recovery is given in the Supplementary Materials (Section 12).
>
> *Decomposability of the joint likelihood into layer-wise summations for tractability. The decomposition is made with few justifications. It would have been more useful if the authors could provide more convincing justifications for this design choice if there were any.*
>
> Thank you for this comment. This design choice is not merely for tractability; it is a core feature motivated by established findings in neurophysiology. As shown by (Yatsenko et al., 2015) and others, neural population covariance naturally decomposes into a sparse, neuron-specific component and a shared, low-rank component modulated by brain state. Our likelihood decomposition is a direct mathematical implication of this principle. Specifically, we provide a formal definition and derivation of likelihood in the Supplementary Materials (Section 9.1), which extends the description from Section 3.3. We have adjusted the main article to emphasize these derivations.
>
> "In short, we model a multilayer network as a product of layer‑specific Gaussian densities whose precision matrices share a joint parameterization in the latent space. We set this up by decomposing the precision matrix of each modality into a sparse component unique to that modality and a shared latent component that captures both the within‑modality and the cross‑modality edges of the multilayer network. This is a more realistic and useful construct, given the experimental evidence cited previously (Yatsenko et al., 2015), and naturally handles heterogeneous $p$ and $n$ across layers."
>
> *While the paper provides a solid solution to the problem at hand, in all honestly, I am unsure whether it fits into the landscape of NeurIPS since the authors solely employ traditional statistical methods and the paper lacks application of neural networks. The authors should enhance the paper's relevancy and impact by demonstrating applications in deep learning contexts to align with contemporary research directions.*
>
> Thank you for raising this point. We respectfully argue that foundational statistical methodology, particularly for graphical models, remains a vibrant and essential part of the NeurIPS landscape. This is reflected in a significant number of recent papers at NeurIPS (Zhang et al., 2024; Negri et al., 2023; Navarro et al., 2024) and many in the broader literature (Roverato and Nguyen, 2024; Tsai et al., 2022; Wang and Allen, 2023; Chang et al., 2022; Kundu et al., 2019).
>
> In addition, our work addresses a critical need in hypothesis-driven scientific fields like neuroimaging, where high-dimensional data and limited sample sizes often make deep learning approaches impractical or prone to overfitting (Wodeyar and Srinivasan, 2022; Smith et al., 2011).  In these data-scarce regimes, theoretically grounded statistical models like multiSLICE are indispensable for generating reliable and interpretable insights.
>
> Far from being disconnected from contemporary research, we see this work as enabling future deep learning applications. By providing a robust method for learning structured, low-dimensional representations from complex multimodal data, our work can provide the basis for more sophisticated deep learning architectures.  We believe this contribution aligns well with NeurIPS's mission of advancing machine-learning methodology, from foundational theory to practical application.
>
> **References**
>
> Andersen Chang, Lili Zheng, and Genevera I Allen. Low-rank covariance completion for graph quilting with applications to functional connectivity. *arXiv preprint arXiv:2209.08273*, 2022.
>
> Suprateek Kundu, Joshua Lukemire, Yikai Wang, and Ying Guo. A novel joint brain network analysis using longitudinal alzheimer’s disease data. *Scientific reports*, 9(1):19589, 2019.
>
> Han Liu, John Lafferty, and Larry Wasserman. The nonparanormal: semiparametric estimation of high dimensional undirected graphs. *Journal of Machine Learning Research*, 10(10), 2009.
>
> Madeline Navarro, Samuel Rey, Andrei Buciulea, Antonio G Marques, and Santiago Segarra. Fair glasso: Estimating fair graphical models with unbiased statistical behavior. *Advances in Neural Information Processing Systems*, 37: 139589–139620, 2024.
>
> Marcello Massimo Negri, Fabricio Arend Torres, and Volker Roth. Conditional matrix flows for gaussian graphical models. Advances in *Neural Information Processing Systems*, 36:25095–25111, 2023.
>
> Alberto Roverato and Dung Ngoc Nguyen. Exploration of the search space of gaussian graphical models for paired data. *Journal of Machine Learning Research*, 25(92):1–41, 2024.
>
> Stephen M Smith, Karla L Miller, Gholamreza Salimi-Khorshidi, Matthew Webster, Christian F Beckmann, Thomas E Nichols, Joseph D Ramsey, and Mark W Woolrich. Network modelling methods for fmri. *Neuroimage*, 54(2): 875–891, 2011
>
> Katherine Tsai, Oluwasanmi Koyejo, and Mladen Kolar. Joint gaussian graphical model estimation: A survey. *Wiley Interdisciplinary Reviews: Computational Statistics*, 14(6):e1582, 2022.
>
> Minjie Wang and Genevera I Allen. Thresholded graphical lasso adjusts for latent variables. *Biometrika*, 110(3):681–697, 2023.
>
> Anirudh Wodeyar and Ramesh Srinivasan. Structural connectome constrained graphical lasso for meg partial coherence. *Network Neuroscience*, 6(4):1219– 1242, 2022.
>
> Eunho Yang, Pradeep Ravikumar, Genevera I Allen, and Zhandong Liu. Graphical models via univariate exponential family distributions. *The Journal of Machine Learning Research*, 16(1):3813–3847, 2015.
>
> Dimitri Yatsenko, Krešimir Josić, Alexander S Ecker, Emmanouil Froudarakis, R James Cotton, and Andreas S Tolias. Improved estimation and interpretation of correlations in neural circuits. *PLoS Computational Biology*, 11(3):e1004083, 2015.
>
> Shengjun Zhang, Xin Fei, Fangfu Liu, Haixu Song, and Yueqi Duan. Gaussian graph network: Learning efficient and generalizable gaussian representations from multi-view images. *Advances in Neural Information Processing Systems*, 37:50361–50380, 2024.

---

> ### Comment · Reviewer_pLq3 · 2025-08-02
>
> Thank you for the responses.
>
> After reading the authors’ rebuttal, I remain unconvinced about the practical usability of the proposed method. I am also working on foundational graphical models, so I acknowledge the importance of the topic and I am not questioning about the relevance of the topic itself.
>
> However, my view is that the literature on graphical models has reached a certain level of maturity. In today’s AI landscape, I believe it would be more impactful to demonstrate how such traditional frameworks can be effectively integrated into modern AI applications. Specifically, I would prefer to see a closer technical connection to contemporary deep learning landscapes, beyond high-level motivation. It could simply be using neural networks to enhance the scalability or modelling expressiveness (like in Zhang et al., 2024) , or allowing us to better handle misspecification or something similar that critically pertains to complex, high-dimensional data.
>
> The authors' experiments on multimodal modeling are interesting, and I had hoped they would offer a compelling case for the applicability of the proposed framework.
>
> While the authors’ claim that *“multiSLIC helps learning structured, low-dimensional representations from complex multimodal data”*, I do not currently see this reflected in the experiments. From my understanding in Lines 978–983, Section 13 of the Supplementary Material, the features for each modality appear to be manually extracted (please correct me if this is not the case).
>
> The paper has a “clean” setup where the total number of variables are well-defined (i.e., the sum of measured features across modalities) and the use of Gaussian assumptions enables the theoretical results to follow naturally. Therefore, in scenarios involving general high-dimensional data, such as images where clearly defined features are not available and misspecification tends to occur, it is unclear how the proposed method aids in representation learning.
>
> I understand that these concerns purely reflect my personal preference for methods that connect more directly to modern applications, and that the authors and the other reviewers may hold different viewpoints. Hence, AC should take these comments lightly.
>
> Regardless, I appreciate the authors’ effort in crafting a technically sound and well-structured paper that may benefit the community. I am willing to consider increasing my score to a borderline accept and the borderline evaluation reflects the concerns outlined above.
>
> Finally, to make it a solid statistical contribution, I would like to see a more explicit discussion of **identifiability of the latent factors from observed data**—especially before making any claims about consistency. While I understand that this may be covered implicitly in the conditions for consistency, and that related results are available in Liu et al. (2017), it would improve the paper’s clarity and completeness.

---

> > ### Author Response · Authors · 2025-08-06
> >
> > We thank the reviewer for providing insightful feedback and an engaging discussion. We hope we have sufficiently addressed the reviewer's comments point by point below.
> >
> > *I believe it would be more impactful to demonstrate how such traditional frameworks can be effectively integrated into modern AI applications.*
> >
> > Thank you for this suggestion. This is certainly an interesting avenue to consider, and so we have adjusted our future works section to incorporate these ideas:
> >
> > "We also consider extending multiSLICE into end-to-end deep-learning workflows by using its outputs—namely the sparse $S$ and low-rank $L$ estimates—to define graph inputs for GNNs. For example, Do et al., 2023 showed that decoupling graph construction (via Graphical LASSO) from GNN training both accelerates convergence ($\approx 40 \%$ faster) and improves performance, while Sriramulu et al., 2023’s ADLNN uses Graphical LASSO as a structural prior before refining edges within a GNN. These single-modality methods underscore how Gaussian graphical models yield efficient, interpretable graphs from multivariate data; multiSLICE generalizes this to a multimodal setting, filling a clear gap.
> >
> > Moreover, multiSLICE can be embedded directly into differentiable architectures. The iterative steps of Algorithm 2 can be unrolled into a fixed-depth module under the OptNet framework (Amos and Kolter, 2017), yielding gradients for multimodal covariance inputs. Alternatively, one can amortize the mapping from sample covariances to ($S, L$) via a small neural network—similar to (Belilovsky et al., 2017)—and feed its outputs into downstream GNNs. Together, these strategies show how multiSLICE can serve both as a modular plug-in and as a fully trainable component for multimodal graph structure learning and downstream prediction."
> >
> > *Features for each modality appear to be manually extracted (please correct me if this is not the case).*
> >
> > Thank you for pointing this out.  We apologize for the overloaded term "features"—in our paper, these are simply the pre-processed measurements from each modality, not learned internal representations.
> >
> > In fact, multiSLICE is can be expressed as learning a mapping
> >
> > $$\mathcal{M}: \lbrace X^{(1)},\dots,X^{(l)} \rbrace \longmapsto
> >   \bigl( \lbrace S \rbrace_{\alpha = 1}^l,L\bigr),$$
> >
> > where $S_{\alpha} \in\mathbb R^{p_{\alpha} \times p_{\alpha}}$ are sparse and modality specific, and,
> >
> > $$L = H H^\top\in\mathbb R^{p\times p}\quad\text{(low–rank)}, \text{ where }  H\in\mathbb R^{p\times r}, r\ll p.$$
> >
> > Here, the columns of \(H\) constitute the low-dimensional, joint, across-modality representation, and $L$ encodes the learned dependency structure both within and between modalities. In this way we make it explicit that multiSLICE takes only the raw measurements $X^{(1)},\dots,X^{(l)}$ as input; As such, Lines 978-983 should instead be understood as referring to these collected measurements, not internal embeddings or hard coded features.
> >
> > *In scenarios involving general high-dimensional data, it is unclear how the proposed method aids in representation learning.*
> >
> > Thank you for this remark. In these more general settings where clearly defined features are not available (e.g.\ raw images, audio, or video), one can precede multiSLICE with a learnable encoder for each modality. For each modality \(\alpha=1,\dots,l\), let
> > $$X_{\alpha}\in \mathbb{R}^{n\times d_{\alpha}},\qquad Z_{\alpha}=f_{\phi}^{(\alpha)}(X_{\alpha})\in \mathbb{R}^{n\times p_{\alpha}},$$
> > and compute its empirical covariance
> > $$\Sigma_{\alpha}=\frac1n\,Z_{\alpha}^T Z_{\alpha}.$$
> > Then, apply
> > $$(S,L)=\mathrm{multiSLICE}(\Sigma_{1},\dots,\Sigma_{l}),$$
> > yielding the sparse $S$ and low-rank adjacency matrices $L$.
> >
> > Assuming the encoded inputs follow the assumptions of our model, as outlined in our Theoretical Analysis, then the all theoretical guarantees carry over unchanged. Then, we suggest two strategies:
> >
> > 1. **Pretrain** each $f_\phi^{(\alpha)}$ (e.g. a CNN) and then run multiSLICE on its outputs, or
> > 2. **Jointly train** $\{f_\phi^{(\alpha)}\}$ and multiSLICE end-to-end by unrolling Algorithm 2 (via Amos and Kolter, 2017) or using an amortized network (like Belilovsky et al., 2017).
> >
> > **References**
> >
> > Brandon Amos and J Zico Kolter. Optnet: Differentiable optimization as a layer in neural networks. In *International conference on machine learning*, pages 136–145. PMLR, 2017.
> >
> > Eugene Belilovsky, Kyle Kastner, Gaël Varoquaux, and Matthew B Blaschko. Learning to discover sparse graphical models. In *International conference on machine learning*, pages 440–448. PMLR, 2017.
> >
> > Ngoc-Dung Do, Truong Son Hy, and Duy Khuong Nguyen. Sparsity exploitation via discovering graphical models in multi-variate time-series forecasting. *arXiv preprint arXiv*:2306.17090, 2023.
> >
> > Abishek Sriramulu, Nicolas Fourrier, and Christoph Bergmeir. Adaptive dependency learning graph neural networks. *Information Sciences*, 625:700–714, 2023.

---

> ### Author Response · Authors · 2025-08-06
>
> We thank you for pointing out the importance of identifiability. Indeed, you are correct to say it is covered implicitly in the conditions for consistency. To make this explicit, we’ve added a short remark in the main text as follows:
>
> "**Remark (Identifiability).**
> By Lemma 6.8 of Liu et al., 2017, the $\Omega / \Omega^T$-isomeric condition is equivalent to requiring the operators
>
> $\mathcal P_{U*}\mathcal P_{\Omega}\mathcal P_{U*}$ and
>
> $\mathcal P_{V*}\mathcal P_{\Omega}\mathcal P_{V*}$
>
> be invertible. Hence these invertibility conditions are exactly the identifiability requirements for parameter matrix $L$."
>
> **References**
>
> Guangcan Liu, Qingshan Liu, and Xiaotong Yuan. A new theory for matrix completion. *Advances in Neural Information Processing Systems*, 30, 2017.

---

> > ### Comment · Reviewer_pLq3 · 2025-08-06
> >
> > Thank you for the detailed responses. Please make sure these discussions are included in the paper, ideally in a dedicated section, so readers in the machine learning community could have clear directions about how the framework can be applied. I will update my score accordingly.

---

> > > ### Author Response · Authors · 2025-08-06
> > >
> > > Thank you for your thoughtful feedback and active engagement with our work—we sincerely appreciate your contribution. As requested, we are preparing a dedicated section summarizing these discussions and offering clear, practitioner-focused guidance on applying our framework. We are also incorporating the other points raised during this discussion period into the manuscript.

---

### Official Review · Reviewer_Wcwi · 2025-06-29

**Clarity:** 3
**Significance:** 1
**Originality:** 2
**Rating:** 5
**Confidence:** 5

**Summary:**

This paper addresses the problem of learning the structure of multi-layer network systems by modeling them as Gaussian graphical models, where network connectivity is encoded in the sparsity pattern of the precision (inverse covariance) matrix. The authors formulate the learning task as a regularized maximum likelihood estimation problem, incorporating sparsity-inducing penalties tailored for multi-layer network structures. They derive sample complexity results that establish the number of samples needed for accurate structure recovery and provide high probability bounds on the estimation error of the precision matrix. Additionally, they propose a numerical optimization algorithm to solve the regularized problem efficiently and validate their approach on two benchmark datasets, demonstrating effective structure recovery.

**Questions:**

1) What is the formal definition of Gaussian distribution on multi-layer networks? If the definition is same as the standard multivariate Gaussian with sparsity pattern of inverse covariance matrix reflecting network structure. Then, the notion of multi-layer network structure is not particularly an interesting probabilistic graphical model.
2) Is it possible to cast this multi-layer network based graphical model in terms of matrix-variate Gaussian distribution? If so, how? If not why?
3) Suggestions for improving simulations were already provided in the weakness section.
4) What is \mathcal{R}(\hat{L})=r$ constraint in Eq (2)? Seems like this quantity is not defined in the paper.

**Ethical Concerns:**

["NO or VERY MINOR ethics concerns only"]

**Final Justification:**

First, the authors did a very detailed and illuminating experiments to support their theoretical results, which was my main concern. Second, they also very clearly addressed the comments of other reviewers. While my initial opinion on the incremental (or textbook exercise) aspect remains the same, nonetheless I've decided to improve the score considering the theoretical analysis, numerical experiments, and the (potential) insights the method suggested for neuroimaging data.

**Limitations:**

The authors, in passing, address as a limitation the fact that their work does not consider the time-varying nature of networks. Personally, I would not consider this as a limitation per se but rather as a natural future avenue. Second, in the checklist, the authors note (paraphrased) that “theorems have assumptions, and those directly state constraints.” I do not completely agree with this viewpoint. Every mathematical theorem has constraints (isn't that the nature of abstract reasoning?). These are not limitations in my view, and hence, I suggest authors to give more serious thought. On societal impact, the authors state there is no potential negative impact. I find this surprising since their simulation work relates to brain studies, where issues such as interpretability, misuse, and privacy often arise.

**Quality:**

2

**Strengths And Weaknesses:**

Strengths:
Overall, the paper is clearly written with technically sound mathematical analyses and derivations. While the results align with expectations from existing graphical model literature, the extension to multi-layer networks is well motivated and fairly presented. The results give insights into the interdependency between variables among layers, sparse edges, and the number of samples, thereby further broadening the statistical learning theory of Gaussian graphical models.

Weaknesses:
The single biggest weakness is that the contribution feels like a “textbook exercise” to existing theory. I use quotes here to emphasize that while the effort is non-trivial, the motivation for this study feels weak, and the results are expected. I could easily imagine giving this problem as a mini project in a class on non-asymptotic statistics for networks. In the forest of sparsity-regularized likelihoods grown over the last two decades, the current value addition is more in highlighting the need for structure learning and then planting the right tree rather than planting a tree and trying to just the need with very laborious math. However, the motivation itself comes across as purely academic extension, and the final dataset results feel like a throw-off – following the standard template of works in this area.

On the technical side, since all norms and the cost function are very well-behaved, and the authors resort to fairly standard proof techniques, it is not clear nor well crafted what the theoretical implications really are. Simulation results do not meaningfully (or qualitatively) justify the derived sample complexity statements. It is also unclear if the considered datasets actually adhered to the hypotheses laid out in the theorems. One suggestion is to include a simple but powerful synthetic experiment, similar to what Martin Wainwright did in his 2009 LASSO paper in IEEE IT. Those experiments were simple but spectacularly teased out theoretical results. Once that type of experimental validation is established, the reader gains credibility in your method before seeing brain or cute-cats-running datasets. This is particularly important here because the authors impose their method on certain datasets without any empirical validation that these datasets demanded the proposed setup.

---

> ### Author Rebuttal · Authors · 2025-07-31
>
> **Weaknesses**
>
> *The contribution feels like a "textbook exercise" to existing theory, the motivation for this study feels weak, and the results are expected.*
>
> Thank you for this comment. We understand that the landscape of sparsity-regularized methodology is saturated. However, our goal is not to add another incremental estimator, but to solve specific scientific problems that current tools leave open.
>
> Neurophysiology studies (e.g., Yatsenko et al., 2015) show that covariances decompose into (i) a *sparse* component, providing undirected associations between brain regions, and (ii) a *low-rank* component, capturing stimuli across modalities. Experimentally, it is of great interest to understand how stimuli influence this low-rank, latent component. Consequently, we adapt this covariance decomposition framework for joint analysis of multiple neuroimaging modalities, and demonstrate its necessity on the Wakeman and Henson (2015) dataset, where existing tools fail to disentangle these structures. We propose the first framework that jointly learns these structures from heterogeneous multimodal data with possibly unequal sample sizes and variable sizes across different data layers. As such, we believe that our method neatly and practically ties together the key areas of multilayer networks, matrix completion, and latent variable Gaussian graphical models.
>
> Furthermore, we provide a key theoretical innovation, a finite‑sample guarantee for joint latent recovery across modalities. In Theorems 5.1 and 5.4, we show that if the latent matrix of each observed modality has the same rank $r$ as the global latent matrix, then all unobserved entries in the full matrix are recovered exactly. Furthermore, we prove that the overall estimation error decomposes cleanly as the sum of the error per layer of each modality. To our knowledge, this is the **first** finite‑sample result for low‑rank completion in a partially‑observed, multilayer setting, effectively bridging the observed and unobserved components of the latent structure.
>
> *Simulation results do not justify the derived sample complexity statements. It is also unclear if the considered datasets actually adhered to the hypotheses laid out in the theorems.*
>
> Thank you for this suggestion. We have added further simulation studies to the Supplementary Materials (Section 12), which are inspired by the experiments from Wainwright (2009). We generate two‑layer Gaussian models with a rank of 2 for the latent component, and vary the dimension ($p = 25, 35, 50$) over a range of sample size ($n = 200, 300, 400, 500, 600$). For each setting, we measure the Frobenius‑norm error of the sparse estimate and the infinity‑norm error of the low‑rank estimate, normalizing both by their values at $n = 200$ to isolate the effect of $p$. We find consistency across different $p$ for both sparse and latent components, validating our non‑asymptotic bounds, even in the challenging regime where many latent entries are unobserved. This experiment demonstrates that our theoretical guarantees hold in practice before application to real neuroimaging data.
>
> *Table 2: Mean scaled norms for sparse and latent estimates as a function of variable dimension \(p\) and sample size \(n\).*
>
> | $$n$$ | $$p=25$$ $$\|\hat S - S^*\|_{F}$$ | $$p=25$$ $$\|\hat L - L^*\|_{\infty}$$ | $$p=35$$ $$\|\hat S - S^*\|_{F}$$ | $$p=35$$ $$\|\hat L - L^*\|_{\infty}$$ | $$p=50$$ $$\|\hat S - S^*\|_{F}$$ | $$p=50$$ $$\|\hat L - L^*\|_{\infty}$$ |
> |:-----|---------------------------------:|--------------------------------------:|---------------------------------:|--------------------------------------:|---------------------------------:|--------------------------------------:|
> | 200  | 1.000                            | 1.000                                 | 1.000                            | 1.000                                 | 1.000                            | 1.000                                 |
> | 300  | 0.882                            | 0.873                                 | 0.885                            | 0.885                                 | 0.888                            | 0.923                                 |
> | 400  | 0.805                            | 0.763                                 | 0.813                            | 0.856                                 | 0.807                            | 0.821                                 |
> | 500  | 0.756                            | 0.748                                 | 0.755                            | 0.785                                 | 0.760                            | 0.744                                 |
> | 600  | 0.703                            | 0.712                                 | 0.702                            | 0.707                                 | 0.713                            | 0.702                                 |
>
> **Questions**
>
> *What is the formal definition of Gaussian distribution on multi-layer networks?*
>
> Thank you for this question. We provide a formal definition and derivation of the likelihood in the Supplementary Materials (Section 9.1) which extends the formalism from Section 3.3. We have modified the main article to emphasize these derivations.
>
> "In short, we model a multilayer network as a product of layer‑specific Gaussian densities whose precision matrices share a joint parameterization in the latent space. We set this up by decomposing the precision matrix of each modality into a sparse component unique to that modality and a shared latent component that captures both the within‑modality and the cross‑modality edges of the multilayer network. This is a more realistic and useful construct, given the experimental evidence cited previously (Yatsenko et al., 2015), and naturally handles heterogeneous $p$ and $n$ across layers."
>
> *Is it possible to cast this model as a matrix-variate Gaussian distribution?*
>
> Thank you for this insightful suggestion. Casting our model in a matrix‑normal framework is indeed an interesting and promising future direction, as it would allow us to capture both sample‑to‑sample and variable‑to‑variable dependencies. Assuming that each modality follows the matrix-variate Gaussian,
>
> $X_\alpha \sim \mathcal{MN}$  $(0,U_\alpha,\Sigma_\alpha)$
>
> we would then impose our sparse + latent decomposition on the column precision as $\Sigma_\alpha^{-1} = S_\alpha + L_\alpha$. A full matrix normal extension could also regularize the row precision matrix $U_\alpha^{-1}$ through a similar split — learning the sample‑to‑sample structure — but we leave this rich extension to future work.
>
> *What is $\mathcal{R}(\hat{L})=r$ constraint in Eq (2)? Seems like this quantity is not defined in the paper.*
>
> The operator $\mathcal R(\cdot)$ denotes matrix rank.  Hence, $\mathcal{R}(\hat{L})=r$ simply enforces that the learned latent matrix $\hat L$ has rank $r$. The operator first appears and is defined in Section 3.1 - the "Notation" portion of "Preliminaries". $r$ is also defined, under Equation 2, which reads "$\rho$ is a tuning parameter for the sparsity in $\hat{S}$ and $r$ is the pre-specified rank of the latent matrix $\hat{L}$." In practice, $r$ serves as a hyperparameter controlling the dimension of the low‑rank subspace shared across layers.
>
> **Limitations**
>
> *In the checklist, the authors should give more serious thought to the limitations.*
>
> We revised the limitations section to:
>
> "We highlight three key regimes in which our model’s guarantees may not hold in practice.  The first is that if measurements from different modalities are gathered simultaneously, our assumption of independent modality measurements is violated. Second, any form of temporal or spatial correlation across observations breaches the i.i.d. requirement underpinning our per‑modality covariance estimators. Lastly, while our sparse, joint‑latent parameterization performs robustly on neuroimaging data, it remains untested in other application areas; practitioners should validate its suitability before applying it to novel domains."
>
> *On societal impact, the authors state there is no potential negative impact.*
>
> Thank you for bringing up this concern. We have adjusted the potential impacts statement:
>
> "Our method provides more accurate and robust estimates of brain connectivity compared to other state-of-the-art methods, which can accelerate fundamental neuroscience research, improve biomarkers for neurological and psychiatric disorders, and ultimately inform better diagnostics and therapies. However, there are potential negative impacts. For one, there is a risk of misinterpretation of estimates. Treating the edges of a correlational network as causal may prompt unsafe interventions. Another concern is privacy. High‐resolution connectomes estimated via multilayer networks can, in principle, carry individual‐specific signatures. Sharing or pooling data without adequate safeguards risks misuse of participants’ brain data. There are also risks in using this method in unintended ways, such as outside clinical or research contexts (e.g., surveillance of cognitive states). Lastly, there are considerations regarding fairness. If the method is applied to heterogeneous populations without proper care, estimates can systematically misrepresent under‑studied groups (e.g., age, ethnicity), leading to biased conclusions."
>
> **References**
>
> Martin J Wainwright. Sharp thresholds for high-dimensional and noisy sparsity recovery using l1-constrained quadratic programming (lasso). *IEEE transactions on information theory*, 55(5):2183–2202, 2009.
>
> Daniel G Wakeman and Richard N Henson. A multi-subject, multi-modal human neuroimaging dataset. *Scientific Data*, 2(1):1–10, 2015.
>
> Dimitri Yatsenko, Krešimir Josić, Alexander S Ecker, Emmanouil Froudarakis, R James Cotton, and Andreas S Tolias. Improved estimation and interpretation of correlations in neural circuits. *PLoS Computational Biology*, 11(3):e1004083, 2015.

---

> > ### Comment · Reviewer_Wcwi · 2025-08-05
> >
> > Thank you for the responses. As I mentioned earlier, and as Reviewer pLq3 aptly put it, "the literature on graphical models has reached a certain level of maturity." Even after reading the authors' response, I remain unconvinced about the extent of theoretical advancement in the structured learning problem considered. I acknowledge the authors’ points regarding practical utility based on certain studies, but it is also well known that the assumptions required for sparsity-regularized estimators are difficult to validate in practice. Finally, the simulations presented focused on infinity norm errors, whereas I was expecting results that quantify the probability of exact recovery versus resampled sample size, as studied in Wainwright (2009). In my view, support recovery metrics are more informative in this context. That said, I appreciate the authors' efforts in running these simulations and in addressing the remaining questions.

---

> > > ### Author Response · Authors · 2025-08-06
> > >
> > > We thank the reviewer for providing insightful feedback and an engaging discussion. We hope we have sufficiently addressed the reviewer's comments point by point below.
> > >
> > > *Even after reading the authors' response, I remain unconvinced about the extent of theoretical advancement in the structured learning problem considered. I acknowledge the authors’ points regarding practical utility based on certain studies, but it is also well known that the assumptions required for sparsity-regularized estimators are difficult to validate in practice.*
> > >
> > > Thank you for raising this point. We note that every high‐dimensional estimator - from sparse, low‐rank, all the way to highly non-linear neural networks - relies on *some set unobservable assumptions* (e.g. sparsity, incoherence, eigenvalue bounds) that cannot be fully verified on finite data, since the true data-generating mechanism is never observed.
> > >
> > > While this makes finite data validation of those assumptions impractical, perhaps a more pragmatic strategy is to assess some desired qualities of the estimated model, such as stability. For example, model selection (via $\rho$ and $r$) in multiSLICE can be easily adapted to study the recovered structures under perturbation, such as with the StARS approach (Liu et al., 2010).
> > >
> > > In our neuroimaging study, we have already demonstrated this enhanced robustness relative to state-of-the-art methods: not only do we observe superior performance in estimating graph topology, but our estimates also exhibit the smallest standard deviations across repetitions (for both sparse and latent components).
> > >
> > > *Finally, the simulations presented focused on infinity norm errors, whereas I was expecting results that quantify the probability of exact recovery versus resampled sample size, as studied in Wainwright (2009). In my view, support recovery metrics are more informative in this context.*
> > >
> > > Thank you for bringing this up. We have updated our simulations to plot recovery probability against the rescaled sample size, while retaining the same norms from our theory, as these norms emphasize magnitude of error, not just support recovery. Specifically:
> > >
> > > - Sparse matrix: we use the entry-wise Frobenius norm $||\hat{S} - S^*||_F$, which penalizes both missing/extra nonzeros and errors in their estimated values.
> > >
> > > - Latent matrix: we use the infinity norm $||\hat{L} - L^*||_\infty$, capturing the worst-case per-row (or per-column) deviation jointly across layers.
> > >
> > > In both cases, we declare a trial "successful" whenever the chosen norm falls below the constant threshold dictated by our theory (Lemma 9.1).
> > >
> > > We have updated the manuscript to include this simulation as well as a set of plots in the Supplementary Materials (which are provided as tables due to limitations of OpenReview):
> > >
> > > Following Wainwright (2009), we evaluate the probability of successful recovery for our estimator via simulation.  Specifically, in each of 100 trials we:
> > > 1. Simulate a two-layer system as detailed in Section 6;
> > > 2. Apply our estimator and compute the relevant error norms;
> > > 3. Declare recovery successful if that norm falls below a constant, as predicted by Lemma 9.1.
> > >
> > > We then plot the empirical success probability (y‐axis) against:
> > > - scaled sample size (right panel), defined as
> > > $$n'_S = \frac{n}{s \log p}$$ and $$n'_L = \frac{n}{C_1 \log p}$$ where $n$ and $p$ are the number of nonzeros and variables in each modality, and $C_1$ is defined in Lemma 9.1.
> > >
> > > As predicted by our theory, once plotted against the appropriately rescaled sample sizes, the error curves from all simulations  align towards a consistent sigmoidal pattern. We observe a strong spike in recovery probability, where the estimator transitions around a critical point (Pr = 0.5), at $n'_S \approx 2.5$ for the sparse estimate, and at $n'_L \approx 2.6e-04$ for the latent estimate.
> > > ''
> > >
> > > We include the tables as a separate comment due to character limitations of each response.
> > >
> > > **References**
> > >
> > > Han Liu, Kathryn Roeder, and Larry Wasserman. Stability approach to regularization selection (stars) for high dimensional graphical models. *Advances in neural information processing systems*, 23, 2010.
> > >
> > > Martin J Wainwright. Sharp thresholds for high-dimensional and noisy sparsity recovery using l1-constrained quadratic programming (lasso). *IEEE transactions on information theory*, 55(5):2183–2202, 2009.

---

> > > > ### Author Response · Authors · 2025-08-06
> > > >
> > > > **Table 1**: Empirical probabilities of successful sparse $S$ and latent $L$ recovery for various sample sizes $n$ and dimensions $p$.
> > > > |  $n$  | $p=25$ $\Pr(\widehat S=S^*)$ | $p=25$ $\Pr(\widehat L=L^*)$ | $p=35$ $\Pr(\widehat S=S^*)$ | $p=35$ $\Pr(\widehat L=L^*)$ | $p=50$ $\Pr(\widehat S=S^*)$ | $p=50$ $\Pr(\widehat L=L^*)$ |
> > > > |:----:|:-------------:|:-------------:|:-------------:|:-------------:|:-------------:|:-------------:|
> > > > | 100  | 0.00 | 0.49 | 0.00 | 0.01 | 0.00 | 0.00 |
> > > > | 200  | 0.06 | 1.00 | 0.00 | 0.49 | 0.00 | 0.03 |
> > > > | 300  | 0.33 | 1.00 | 0.00 | 0.91 | 0.00 | 0.14 |
> > > > | 400  | 0.75 | 1.00 | 0.00 | 1.00 | 0.00 | 0.19 |
> > > > | 500  | 0.97 | 1.00 | 0.20 | 1.00 | 0.00 | 0.27 |
> > > > | 600  | 1.00 | 1.00 | 0.61 | 1.00 | 0.00 | 0.25 |
> > > > | 700  | 1.00 | 1.00 | 0.85 | 1.00 | 0.02 | 0.36 |
> > > > | 800  | 1.00 | 1.00 | 0.97 | 1.00 | 0.10 | 0.44 |
> > > > | 900  | 1.00 | 1.00 | 1.00 | 1.00 | 0.33 | 0.54 |
> > > > | 1000 | 1.00 | 1.00 | 1.00 | 1.00 | 0.71 | 0.62 |
> > > > | 1100 | 1.00 | 1.00 | 1.00 | 1.00 | 0.90 | 0.67 |
> > > > | 1200 | 1.00 | 1.00 | 1.00 | 1.00 | 0.98 | 0.63 |
> > > > | 1300 | 0.99 | 1.00 | 1.00 | 1.00 | 1.00 | 0.72 |
> > > > | 1400 | 0.99 | 1.00 | 1.00 | 1.00 | 1.00 | 0.75 |
> > > > | 1500 | 0.99 | 1.00 | 1.00 | 1.00 | 1.00 | 0.80 |
> > > >
> > > > **Table 2**: Empirical sparse $S$ recovery probability as a function of the scaled sample size $n'_S$, shown separately for dimensions \(p=25,35,50\)
> > > > | $p=25$ $n'_S$ | $p=25$ $\Pr(\widehat S=S^*)$ | $p=35$ $n'_S$ | $p=35$ $\Pr(\widehat S=S^*)$ | $p=50$ $n'_S$ | $p=50$ $\Pr(\widehat S=S^*)$ |
> > > > |:-:|:-:|:-:|:-:|:-:|:-:|
> > > > |0.66|0.00|0.420|0.00|0.264|0.00|
> > > > |1.322|0.06|0.840|0.00|0.527|0.00|
> > > > |1.983|0.33|1.259|0.00|0.791|0.00|
> > > > |2.644|0.75|1.679|0.00|1.054|0.00|
> > > > |3.305|0.97|2.099|0.20|1.318|0.00|
> > > > |3.966|1.00|2.519|0.61|1.581|0.00|
> > > > |4.627|1.00|2.939|0.85|1.845|0.02|
> > > > |5.288|1.00|3.358|0.97|2.108|0.10|
> > > > |5.949|1.00|3.778|1.00|2.372|0.33|
> > > > |6.610|1.00|4.198|1.00|2.635|0.71|
> > > > |7.271|1.00|4.618|1.00|2.899|0.90|
> > > > |7.932|1.00|5.038|1.00|3.162|0.98|
> > > > |8.593|0.99|5.457|1.00|3.426|1.00|
> > > > |9.254|0.99|5.877|1.00|3.689|1.00|
> > > > |9.915|0.99|6.297|1.00|3.953|1.00|
> > > >
> > > > **Table 3**: Empirical latent $L$ recovery probability as a function of the scaled sample size $n'_L$, shown separately for dimensions \(p=25,35,50\)
> > > > |$p=25$ $n'_L$|$p=25$ $\Pr(\widehat L=L^*)$|$p=35$ $n'_L$|$p=35$ $\Pr(\widehat L=L^*)$|$p=50$ $n'_L$|$p=50$ $\Pr(\widehat L=L^*)$|
> > > > |:-:|:-:|:-:|:-:|:-:|:-:|
> > > > |3.36e-04|0.49|1.05e-04|0.01|2.97e-05|0.00|
> > > > |6.72e-04|1.00|2.09e-04|0.49|5.94e-05|0.03|
> > > > |1.01e-03|1.00|3.14e-04|0.91|8.91e-05|0.14|
> > > > |1.34e-03|1.00|4.18e-04|1.00|1.19e-04|0.19|
> > > > |1.68e-03|1.00|5.23e-04|1.00|1.48e-04|0.27|
> > > > |2.02e-03|1.00|6.28e-04|1.00|1.78e-04|0.25|
> > > > |2.35e-03|1.00|7.32e-04|1.00|2.08e-04|0.36|
> > > > |2.69e-03|1.00|8.37e-04|1.00|2.38e-04|0.44|
> > > > |3.02e-03|1.00|9.41e-04|1.00|2.67e-04|0.54|
> > > > |3.36e-03|1.00|1.05e-03|1.00|2.97e-04|0.62|
> > > > |3.70e-03|1.00|1.15e-03|1.00|3.27e-04|0.67|
> > > > |4.03e-03|1.00|1.26e-03|1.00|3.56e-04|0.63|
> > > > |4.37e-03|1.00|1.36e-03|1.00|3.86e-04|0.72|
> > > > |4.71e-03|1.00|1.46e-03|1.00|4.16e-04|0.75|
> > > > |5.04e-03|1.00|1.57e-03|1.00|4.45e-04|0.80|

---

> > > > > ### Author Response · Authors · 2025-08-07
> > > > >
> > > > > We appreciate the time you’ve already devoted to reviewing our paper.
> > > > >
> > > > > We have responded to each of your earlier points in the discussion thread. If you have any follow-up questions—or if anything remains unclear—please let us know. The discussion phase closes tomorrow, and we’d be happy to clarify anything before then.
> > > > >
> > > > > Thank you again for your valuable feedback.

---

> > > > > > ### Comment · Reviewer_Wcwi · 2025-08-08
> > > > > >
> > > > > > Thank you once again for addressing my comments and presenting a detailed simulation results. Please include these results (at least some snippets) in the revised paper. Based on the evaluation of other reviewers and the authors responses through out the discussion period, I've decided to increase my score.
> > > > > >
> > > > > > Two clarifications*
> > > > > > - My initial opinion on the practical utility and the textbook extension remains the same. Since I firmly believe it is the duty of theorists -- whether in service of science or engineering -- to formulate assumptions or mathematical models grounded in data, I cannot accept the authors’ argument that "....[our method] cannot be fully verified on finite data, since the true data-generating mechanism is never observed."
> > > > > > - Support recovery results presented by the authors justify Thm 5.4 (in a rather circular way) but not Thm 5.3, which is what wainwright (2009) paper presents. I'm guessing that the authors are aware of this important distinction. I also understand why the authors chose the circular way. In general numerical demonstration of support recovery results are hard, except for simple cases like in sparse linear models.
> > > > > >
> > > > > > *Although I know my views may not matter much once the score is improved :), I still feel it is my responsibility to share them.

---

> > > > > > > ### Author Response · Authors · 2025-08-08
> > > > > > >
> > > > > > > We sincerely thank the reviewer for the constructive feedback, active engagement during the discussion, as well as these final clarifications. We greatly value these insights and are actively incorporating all the suggested edits into the revised manuscript.

---

### Official Review · Reviewer_JX1i · 2025-07-01

**Clarity:** 3
**Significance:** 3
**Originality:** 3
**Rating:** 5
**Confidence:** 1

**Summary:**

This paper presents multiSLICE, a novel framework for learning multilayer Gaussian graphical models where each layer can have a different number of variables and samples. It models each layer’s precision matrix as a sum of a sparse matrix (intralayer structure) and a shared low-rank matrix (interlayer structure). The two-stage algorithm estimates sparse and latent components per layer, then combines them using a matrix completion approach. The method is theoretically grounded, handles unaligned data, and outperforms prior methods on both simulated and neuroimaging data.

**Questions:**

Please describe how hyper-parameters are selected

**Ethical Concerns:**

["NO or VERY MINOR ethics concerns only"]

**Final Justification:**

I appreciate the author’s response, which addresses my concern. I continue to uphold my previous rating, leaning toward accepting this paper.

**Limitations:**

Yes

**Quality:**

3

**Strengths And Weaknesses:**

Strengths:

1. Addresses a significant challenge in heterogeneous multilayer graph learning. The method handles differing numbers of variables and samples across layers—an important and previously under-addressed problem. This makes it widely applicable to real-world multimodal settings, such as neuroscience or genomics.
2. Novel integration of sparse and low-rank modeling with matrix completion. While individually inspired by existing techniques, the combination of latent GGMs, sparse precision estimation, and low-rank matrix completion is original in this context. The model provides both within-layer network structure and explicit interlayer edge estimation.
3. Scalable, theoretically grounded algorithm. The two-stage procedure (SLICE + matrix assembly) is computationally tractable and supported by recovery guarantees under reasonable assumptions. The design allows each layer to be processed separately, enabling parallelism and better scalability to many layers.
4. Strong empirical performance across tasks. MultiSLICE consistently outperforms baselines in recovering both sparse and latent structure in simulations. On real neuroimaging data, it produces networks with higher modularity and lower latent entropy, indicating more meaningful and interpretable structure.

Weaknesses:


The paper does not discuss how to choose key parameters like the latent rank or sparsity penalties in practice. This limits immediate usability and may affect performance if the method is misconfigured.
Incomplete comparisons with recent baselines.

---

> ### Author Rebuttal · Authors · 2025-07-31
>
> **Weaknesses**
>
> *The paper does not discuss how to choose key parameters like the latent rank or sparsity penalties in practice. This limits immediate usability and may affect performance if the method is misconfigured.*
>
> We appreciate the reviewer highlighting this point. A full description of our cross-validation procedure for tuning hyperparameters is given in the Supplementary Materials (Section 11.1).  We have also adjusted the main article, specifically Sections 6 and 7, to better describe our procedure, which is a 3-fold cross-validation over a grid of $\rho$ and $r$. Hyperparameters are selected as the combination with the best held-out log-likelihood. This is also implemented in our codebase on the anonymous GitHub repository as the `cv.multislice(Xs, folds, rhos, rs)` function, where `Xs`, `folds`, `rhos`, and `rs` are the input data, number of folds for cross-validation, vector of candidate $\rho$ values, and vector of candidate $r$ values, respectively. A set of simulation studies looking at the effects of $\rho$ and $r$ on parameter recovery is given in the  Supplementary Materials (Section 12).
>
> *Incomplete comparisons with recent baselines.*
>
> Thank you for this comment. In Section 2, we review eight recent state‑of‑the‑art multilayer and latent‑factor methods, and in Sections 6–7 compare multiSLICE with those methods. Given some of the design differences between methods, such as sample size ($n$) and the number of variables ($p$), we are unable to compare all of them to multiSLICE. If a method does not assume a latent structure, we were unable to compare our latent estimate from multiSLICE. In the real data analysis, because some methods do not posit a latent factor model or require equal sample sizes across layers, direct application to the dataset falls outside their scope. If there are other baselines you think we are missing, we would be happy to include them.
>
> **Questions**
>
> *Please describe how hyper-parameters are selected*
>
> Please see our response to the first weakness.

---

### Official Review · Reviewer_ioDz · 2025-07-02

**Clarity:** 3
**Significance:** 2
**Originality:** 3
**Rating:** 5
**Confidence:** 2

**Summary:**

This paper introduces multiSLICE, a latent-variable Gaussian graphical model tailored for multilayer networks. The key idea is to decompose each layer’s precision matrix into a sparse intra-layer component plus a shared low-rank component spanning all layers. This low-rank latent factor effectively captures inter-layer connections (influences common across modalities) while the sparse part captures layer-specific associations (e.g. brain-region connections within each modality). Notably, multiSLICE can handle heterogeneous graph structures, without forcing resampling or alignment. The authors propose an efficient two-stage algorithm: first, estimate each layer’s sparse + low-rank structure separately via an EM-like procedure (Algorithm 1), then latent components via a block-wise SVD matrix completion step (Algorithm 2). Finally, the paper empirically validates multiSLICE on both synthetic data (varying number of layers and latent rank) and real multimodal neuroimaging data (MEG + fMRI + sMRI), showing improved accuracy over several state-of-the-art multilayer GGM baselines. Proofs are provided in the supplementary.

**Questions:**

How might the multiSLICE framework be adapted for time-varying or longitudinal multilayer data?

It would be insightful to see how multiSLICE performs in a scenario where those methods are applicable. For example, could you run an additional simulation where all layers have equal n and p, and compare multiSLICE with baselines in that controlled setting? This would show that multiSLICE is not only superior in heterogeneous regimes but also competitive in homogeneous ones.

The method introduces a rank parameter r for L. How should these be chosen in practice? providing guidance or an automatic strategy (even in supplementary) for selecting these hyperparameters would make the method easier to use.

**Ethical Concerns:**

["NO or VERY MINOR ethics concerns only"]

**Final Justification:**

The response is informative, and it has addressed most of my concerns.

Please consider to include these additional discussions in the paper.

**Limitations:**

yes

**Quality:**

3

**Strengths And Weaknesses:**

Strengths:

-	Combines sparse + low-rank modeling in a multilayer graphical model formulation.
-	Authors provide a comprehensive theoretical background.
-	Empirical results (synthetic and real neuroimaging data) show superior performance vs. strong baselines.

Weaknesses:
-	Notation and matrix completion step could be explained with more intuition and clearly, especially in Algorithm 2, where the authors use block-wise SVD to combine latent components from different layers. It would be helpful to elaborate how this procedure reconstructs a coherent global low-rank matrix from layer-wise estimates and how it addresses issues arising from partial observability or node mismatch across layers.

---

> ### Author Rebuttal · Authors · 2025-07-31
>
> **Weaknesses**
>
> *Notation and matrix completion step could be explained with more intuition and clearly, especially in Algorithm 2, where the authors use block-wise SVD to combine latent components from different layers. It would be helpful to elaborate how this procedure reconstructs a coherent global low-rank matrix from layer-wise estimates and how it addresses issues arising from partial observability or node mismatch across layers.*
>
> Thank you for this comment. We assume disjoint node sets across layers, consistent with standard multimodal neuroimaging collection (e.g., no one‑to‑one mapping between EEG channels and fMRI voxels). This motivates the matrix completion step and matches real‑world data, where measurements in modality A and modality B rarely align exactly.
>
> The matrix completion step in Algorithm 2 follows the block SVD approach from Bishop and Yu (2014). From preliminary simulation studies, comparing block‑SVD against nuclear‑norm penalization and other popular completion methods, we found that this approach achieved the best recovery of missing entries. This also mirrors the results observed in Chang et al. (2022), who found that covariances were more accurately recovered in simulation studies using this method compared to others. To maintain focus, we defer a more exhaustive comparison of alternative approaches to future work.
>
> We have expanded the explanation of Algorithm 2 to make the intuition more explicit. ``For the matrix completion portion, Algorithm 2 first computes per‑layer SVDs,
>
> $L_i = U_i\,\Lambda_i\,U_i^T,\quad i=1,\dots,l,$
>
> then ``intersects'' each pair of latent subspaces by forming
>
> $L_{ij}=U_i\,\Lambda_i^{1/2}\,\Lambda_j^{1/2}\,U_j^T,$
>
> and finally assembles all blocks $\{L_{ij}\}$ in the global low‑rank matrix, $L$. As an illustrative case, we can consider a two-layer case, where we reconstruct $L_{12}=U_1\Lambda_1^{1/2}\Lambda_2^{1/2}U_2^T$ before assembling them into $L$. This block‑SVD fusion both preserves the latent factors of each layer exactly and aligns their shared structure. As formalized in Theorem 5.1, if the latent matrix of all observed layers has a rank equal to the global rank, then all inter‑layer blocks and hence the full matrix are recovered exactly under our disjoint‑node setup.''
>
> **Questions**
>
> *How might the multiSLICE framework be adapted for time-varying or longitudinal multilayer data?*
>
> Thank you for this insightful question. We have added a new discussion in the Conclusion outlining extensions of multiSLICE to time‑varying or longitudinal settings:
>
> ``To adapt multiSLICE to a time-varying or longitudinal multilayer data, we suggest the following approaches.
>
> Sliding‑windows: Apply multiSLICE over overlapping time windows, yielding a time-varying latent subspace. Although simple and widely used in neuroimaging, this approach may undo some preprocessing steps taken in neuroimaging data analysis (Lindquist, 2024).
>
> Change-point detection: Track a summary statistic of the joint latent factors, such as the principal angles between successive $L$ matrices, and employ offline or online change‑point methods (Cribben et al., 2012; Chen and Zhang, 2015) to segment the time series. Apply multiSLICE separately within each stationary segment.
>
> Temporal regularization: Reformulate the objective function to jointly estimate ${L^{(t)}}_{t=1}^T$, with penalties enforcing smoothness or sparsity in time as in Hallac et al. (2017).
>
> For strictly repeated (longitudinal) measurements, one only needs to treat each time point as a separate ``layer'' in multiSLICE, ordering the covariance inputs by time so that the inter‑block estimates $\{L_{t,t+1}\}$ capture the within‑subject evolution directly.''
>
> *It would be insightful to see how multiSLICE performs in a scenario where those methods are applicable. For example, could you run an additional simulation where all layers have equal n and p, and compare multiSLICE with baselines in that controlled setting? This would show that multiSLICE is not only superior in heterogeneous regimes but also competitive in homogeneous ones.*
>
> Thank you for this suggestion. We have added further simulation studies to the Supplementary Materials (Section 12) that compare multiSLICE to the baselines in the controlled setting you have described. In particular, we utilize a two-layer system where $\mathcal{R}(L^*) = 2$. We then vary $p$ and $n$ equally to compare multiSLICE with BANS, JMMLE, LRGQ, and MLGGM in a homogeneous setting. For brevity, we exclude CNJGL, BJEMGM, CFR, and coglasso, since neither the F1 score nor the $\sin\theta$ improved with increasing $n$. The results are outlined in the table below, where each value is the average of 100 replications. We find that multiSLICE, except for the F1 score with $p, n = 50$, has a superior performance over the state-of-the-art methods by a wide margin in this challenging simulation setting.
>
> *Table 1: F1 and* $\sin\theta$ *results for various model and sample sizes.*
>
> | $p, n$ | BANS F1    | BANS $\sin\theta$ | JMMLE F1   | JMMLE $\sin\theta$ | LRGQ F1    | LRGQ $\sin\theta$ | MLGGM F1   | MLGGM $\sin\theta$ | multiSLICE F1 | multiSLICE $\sin\theta$ |
> |:-----|-----------:|------------------:|-----------:|-------------------:|-----------:|------------------:|-----------:|-------------------:|--------------:|------------------------:|
> | 50   | **0.058**  |           0.999   | 0.000      |           0.998   | 0.000      |           1.00    | 0.033      |           0.998   | 0.049         | **0.994**              |
> | 100  | 0.049      |           0.995   | 0.000      |           0.998   | 0.000      |           1.00    | 0.052      |           0.998   | **0.089**     | **0.929**              |
> | 200  | 0.065      |           0.996   | 0.000      |           0.998   | 0.000      |           1.00    | 0.092      |           0.997   | **0.173**     | **0.501**              |
> | 300  | 0.061      |           0.996   | 0.014      |           0.997   | 0.073      |           1.00    | 0.138      |           0.997   | **0.255**     | **0.225**              |
>
> *The method introduces a rank parameter r for L. How should these be chosen in practice? providing guidance or an automatic strategy (even in supplementary) for selecting these hyperparameters would make the method easier to use.*
>
> We appreciate the reviewer highlighting this point. A full description of our cross-validation procedure for tuning hyperparameters is in the Supplementary Materials (Section 11.1). We have also adjusted the main article, specifically Sections 6 and 7, to better describe our procedure, which is a 3-fold cross-validation over a grid of $\rho$ and $r$. Hyperparameters are selected as the combination with the best held-out log-likelihood. This is also implemented in our codebase on the anonymous GitHub repository as the `cv.multislice(Xs, folds, rhos, rs)` function, where `Xs`, `folds`, `rhos`, and `rs` are the input data, number of folds for cross-validation, vector of candidate $\rho$ values, and vector of candidate $r$ values, respectively. A set of simulation studies looking at the effects of $\rho$ and $r$ on parameter recovery is given in the Supplementary Materials (Section 12).
>
> **References**
>
> William E Bishop and Byron M Yu. Deterministic symmetric positive semidefinite matrix completion. *Advances in Neural Information Processing Systems*, 27, 2014.
>
> Andersen Chang, Lili Zheng, and Genevera I Allen. Low-rank covariance completion for graph quilting with applications to functional connectivity. *arXiv preprint arXiv:2209.08273*, 2022.
>
> Hao Chen and Nancy Zhang. Graph-based change-point detection. *The Annals of Statistics*, 43(1):139 – 176, 2015
>
> Ivor Cribben, Ragnheidur Haraldsdottir, Lauren Y Atlas, Tor D Wager, and Martin A Lindquist. Dynamic connectivity regression: determining state related changes in brain connectivity. *Neuroimage*, 61(4):907–920, 2012.
>
> David Hallac, Youngsuk Park, Stephen Boyd, and Jure Leskovec. Network inference via the time-varying graphical lasso. In *Proceedings of the 23rd ACM SIGKDD international conference on knowledge discovery and data mining*, pages 205–213, 2017.
>
> Martin A Lindquist. Sliding windows analysis can undo the effects of preprocessing when applied to fmri data. *bioRxiv*, pages 2023–10, 2024.

---

> > ### Author Response · Authors · 2025-08-07
> >
> > We appreciate the time you’ve already devoted to reviewing our paper.
> >
> > We have responded to each of your earlier points in the discussion thread. If you have any follow-up questions—or if anything remains unclear—please let us know. The discussion phase closes tomorrow, and we’d be happy to clarify anything before then.
> >
> > Thank you again for your valuable feedback.

---

> > ### Comment · Reviewer_ioDz · 2025-08-08
> > **Thank you authors for the informative response**
> >
> > Thank you authors for the response. My concerns have been mostly resolved.

---

### Author Response · Authors · 2025-08-09

As the discussion period comes to a close, we thank the reviewers for their time, thoughtful feedback, and constructive discussion. In response, we made the following key improvements to the manuscript, among others:

1. **Clarified Algorithm 2 and matrix completion**: Expanded the explanation with more intuition, an illustrative case, and formal alignment to the disjoint-node setup.

2. **Additional simulations**:
Added a homogeneous-layer experiment to show improvements over the state-of-the-art when all layers have equal $p$ and $n$.
Included Wainwright 2009 style sample complexity validations and probability-of-recovery analyses, showcasing how the results align with our theory.

3. **Detailed hyperparameter selection**: Emphasized and expanded our description (in main text and supplementary) of the cross-validation procedure for selecting $\rho$ and $r$ without *a priori* knowledge.

4. **Addressed theoretical and practical concerns**: Clarified the formal definition of Gaussian multilayer networks, the rank constraint in Eq. (2), provided a remark for the identifiability of the model, and discussed a potential matrix-normal extension.

5. **Integrated into modern AI workflows**: Expanded the future work section to outline how multiSLICE’s sparse and low-rank estimates could serve as graph inputs for GNNs or be embedded as differentiable modules in end-to-end architectures, bridging traditional graphical models with deep learning methods.

6. **Expanded limitations and societal impact**: Detailed scenarios where guarantees may not hold, potential misuse risks, privacy concerns, and fairness considerations.

We appreciate the reviewers’ engagement, which has helped strengthen both the clarity and rigor of our work.

---

### Note · Authors · 2025-08-15

We would like to reiterate and expand upon the key improvements made in response to the reviewers’ thoughtful and constructive feedback. We thank the reviewers, the area chairs, and all involved for their engagement, which has significantly strengthened the clarity, rigor, and broader impact of our work.

**Clarified model setup and notation**: Refined the presentation of the Gaussian multilayer network model, including the decomposition into modality-specific sparse parameters and joint latent parameters, and bolstered the underlying motivation for this formulation.

**Clarified Algorithm 2 and matrix completion**: Expanded the explanation with more intuition, an illustrative case, and formal alignment to the disjoint-node setup.

**Enhanced theoretical motivation**: Strengthened the rationale for our theoretical analysis, particularly the choice of norms, and provided an explicit remark on identifiability. Clarified the formal definition of Gaussian multilayer networks, the rank constraint in Eq. (2), and discussed a potential matrix-normal extension.

**Additional simulations**: Added a homogeneous-layer experiment demonstrating improvements over the state-of-the-art when all layers have equal $p$ and $r$. Included Wainwright (2009)-style probability-of-recovery analyses, and corresponding definitions for scaled sample sizes. Added new figures and descriptive explanations to illustrate these results.

**Detailed hyperparameter selection**: Expanded the description (in both main text and supplementary) of the cross-validation procedure for selecting $\rho$ and $r$ without a priori knowledge.

**Integrated into modern AI workflows**: Expanded the future work section to outline how multiSLICE’s sparse and low-rank estimates could serve as graph inputs for GNNs or be embedded as differentiable modules in end-to-end architectures, bridging traditional graphical models with deep learning methods.

**Expanded limitations and societal impact**: Detailed scenarios where our guarantees may not hold, potential misuse risks, privacy concerns, and fairness considerations.

**General clarity and flow improvements**: Made targeted edits throughout for readability and precision, including avoiding overloaded terminology (e.g., replacing “features” in the supplementary materials).

We are grateful for the reviewers’ insightful suggestions, which have been instrumental in refining both the technical content and presentation of the manuscript.

---

### Decision · Program_Chairs · 2025-09-17

**Decision:**

Accept (poster)

**Comment:**

The paper proposes multiSLICE, a latent-variable Gaussian graphical model for multilayer networks, combining sparse within-layer structure with a shared low-rank latent component.

Reviewers’ main concerns about clarity, theory - experiment alignment, and practical usability have been addressed.

For the camera-ready, the authors should (1) integrate the added simulation results and probability-of-recovery plots, (2) include the identifiability discussion explicitly, (3) expand the guidance on hyperparameter tuning and practical usage, and (4) add the discussion on links to deep learning and broader applications.